# A Hybrid Machine Learning Approach to Screen Optimal Predictors for the Classification of Primary Breast Tumors from Gene Expression Microarray Data

**DOI:** 10.3390/diagnostics13040708

**Published:** 2023-02-13

**Authors:** Nashwan Alromema, Asif Hassan Syed, Tabrej Khan

**Affiliations:** 1Department of Computer Science, Faculty of Computing and Information Technology Rabigh (FCITR), King Abdulaziz University, Jeddah 22254, Saudi Arabia; 2Department of Information Systems, Faculty of Computing and Information Technology Rabigh (FCITR), King Abdulaziz University, Jeddah 22254, Saudi Arabia

**Keywords:** primary breast tumor, gene-biomarkers, hybrid-feature selection approach, filter-based fs, two-tailed unpaired *t*-test, meta-heuristics techniques, supervised machine learning classifiers, breast tumor prediction

## Abstract

The high dimensionality and sparsity of the microarray gene expression data make it challenging to analyze and screen the optimal subset of genes as predictors of breast cancer (BC). The authors in the present study propose a novel hybrid Feature Selection (FS) sequential framework involving minimum Redundancy-Maximum Relevance (mRMR), a two-tailed unpaired *t*-test, and meta-heuristics to screen the most optimal set of gene biomarkers as predictors for BC. The proposed framework identified a set of three most optimal gene biomarkers, namely, MAPK 1, APOBEC3B, and ENAH. In addition, the state-of-the-art supervised Machine Learning (ML) algorithms, namely Support Vector Machine (SVM), K-Nearest Neighbors (KNN), Neural Net (NN), Naïve Bayes (NB), Decision Tree (DT), eXtreme Gradient Boosting (XGBoost), and Logistic Regression (LR) were used to test the predictive capability of the selected gene biomarkers and select the most effective breast cancer diagnostic model with higher values of performance matrices. Our study found that the XGBoost-based model was the superior performer with an accuracy of 0.976 ± 0.027, an F1-Score of 0.974 ± 0.030, and an AUC value of 0.961 ± 0.035 when tested on an independent test dataset. The screened gene biomarkers-based classification system efficiently detects primary breast tumors from normal breast samples.

## 1. Introduction

In recent years despite the technological advances in imaging tools, the earlier detection of BC has remained a tenacious challenge. According to a recent statistical analysis, BC is the most predominant cancer in women and the second most common cause of mortality in undeveloped and developed countries (https://www.who.int/news-room/fact-sheets/detail/breast-cancer/, accessed on 10 October 2022). An earlier cancer diagnosis reduces the probability of death in cancer patients, which can be achieved using comprehensive screening programs [1,2]. Moreover, understanding the underlying mechanism and pathogenesis would improve BC’s efficient diagnosis and treatment. BC’s formation and development involve genomic, transcriptomic, and epigenomic factors [3]. Therefore, understanding the pathogenesis of cancer from a molecular perspective will contribute toward our goal of the early detection and effective treatment of BC. Conventional techniques such as histopathological classification and imaging tools, such as ultrasound, mammography, and magnetic resonance imaging, have been proven beneficial [4,5,6]. However, they offer little information about the mechanism of cancer development and progression [7,8].

On the contrary, with the recent development and advancement in DNA microarray technology, large-scale genomic transcriptomic expression data archives, namely the Cancer Genome Atlas [9] and CuMiDa [10], have been developed. These provide a platform to explore and understand the pathogenesis of cancer formation and proliferation from a molecular perspective. Analyzing the expression of genes among newly diagnosed BC patients [11,12,13,14,15] and those undergoing treatment [16,17] provides a better understanding of the disease progression and prognosis. Large-scale cancer genomics screening programs explore novel BC gene biomarkers to improve early detection and reduce mortality. It is essential to systematically analyze the possible effect of individual genes, or a combination of genes, as gene biomarkers indicative of BC to discover more potential predictors to aid early detection.

However, evaluating each gene or a combination of genes can be viewed as a Feature Selection (FS) problem executed on a high-throughput multi-dimensional microarray gene expression dataset. The number of features (genes) is significantly greater than the number of instances in the high throughput gene expression microarray dataset. Thus, many features lead to overfitting problems (the curse of dimensionality) where the standard ML algorithms overfit the data, leading to a significant performance difference between the trained and tested supervised classifier-based predictive models [18,19]. Therefore, to address the problem, feature (gene) selection techniques have been applied to BC microarray gene expression data to reduce the curse of dimensionality by eliminating redundant and non-informative features (genes) [20,21].

Typically, FS approaches can be grouped into four groups, namely, (1) filter, (2) wrapper, (3) embedded, and (4) hybrid methods [22]. Filter methods utilize statistical properties, namely scores in various statistical tests, to determine the correlation of the particular feature with the response variable. Moreover, selecting the most statistically relevant feature in the filter method is independent of any machine learning algorithms and collinearity between the features. Wrapper methods use learning methods to search for optimal subsets of features. The selection of the best-performing subset of features in a wrapper method is based on the performance of the specific classifier we are trying to fit on a given dataset. The Wrapper methods typically apply metaheuristics search approaches to evaluate all the possible combinations of features against the evaluation metrics, and these metaheuristics-based wrapper methods have shown excellent performance. However, wrapper approaches are computationally expensive because they employ a classifier to evaluate each subset of features. In embedded methods, the FS process is typically built-in with the classifier to determine the optimal feature subset. Hybrid methods combine filter and wrapper approaches, and the hybrid methods of FS are popularly used by researchers worldwide. The Hybrid FS approach uses the advantages of the filter and wrapper FS techniques [23,24,25,26,27,28,29,30,31,32,33].

Feature selection in a high ratio of features to samples in microarray gene expression data is an example of an NP-hard problem. Therefore, heuristic-based global minima search algorithms are most appropriate for finding the most optimal solutions in these complicated NP-hard problems. Moreover, the hybrid FS methods are sometimes more applicable than filter-based methods in screening gene biomarkers from the cancer microarray gene expression dataset [20]. The nature-inspired FS methods select the best optimal feature subset using heuristic search to maximize the classification accuracy in binary and multi-class classification problems [34]. Metaheuristics algorithms have also been used to solve many NP-hard problems in various fields, such as function optimization [35,36,37], feature extraction for the image-based classification of cancer [38], feature selection for cancer diagnosis [39,40], and biomedical engineering [41,42,43,44] and circuit design [45,46].

Motivation: FS studies applying metaheuristics in the hybrid learning approach for gene biomarker selection are still limited in BC classification, thereby enabling an earlier diagnosis of BC. Thus, the impact studies involving the hybrid approaches using heuristics algorithms require further research. These facts motivated us to propose a new framework involving a systematic, comprehensive analysis of the other hybrid FS methods, followed by various classifier-based primary tumor prediction analyses to screen genes that have robust diagnostic efficiency in detecting the earlier stages of BC using the microarray gene expression dataset. Moreover, our study also provides biological insight into the selected subset of features. Thus, our method aims to determine genes with robust diagnostic capability while also providing biological insights that would help understand the early molecular mechanisms of BC from the topological and physical aspects.

Contribution: The main contributions and the novel aspects of the present study are the following:
In the present study, we propose a new framework of a hybrid feature selection approach to screen predictors from microarray gene expression data with better diagnostic efficacy in predicting the earlier stages of BC. In this context, our proposed hybrid-based FS approach consists of two steps.
The first stage involves systematically applying the filter-based, statistical, and metaheuristics optimization approaches to select the best subset of the predictor.The second stage consists of the classification stage. In this stage, the screened subset of genes from the FS stage was used to identify BC features with the highest classification accuracy.We compared our proposed hybrid FS method with the state-of-the-art techniques developed in recent years.The experimental results showed that the hybrid FS method presented in this study is robust in classifying primary BC tumors into binary categories (normal and primary breast tumor classes), outperforming the state-of-the-art hybrid FS methods developed to screen gene biomarkers for an earlier diagnosis of BC.

In the present study, a machine learning-based hybrid FS framework was developed to screen the best optimal feature subset to build a classification model for earlier detection of primary BC, as represented in Figure 1. Our proposed supervised classification model was created using the curated gene expression microarray data of 176 primary breast cancer patients and 10 normal breast samples. First, the gene expression data Breast_GSE22820 [47] obtained from the CuMiDa database [10] is preprocessed. The preprocessed data were then introduced into a hybrid FS framework that involves a sequential implementation of filter-based, statistical, and five metaheuristics optimization approaches.

The hybrid FS method was designed to screen a stable and relevant subset of stable gene biomarkers (features) to classify primary breast cancer samples from normal breast samples. The hybrid FS approach yielded the five most optimal subsets of features. First, five subsets of the gene microarray data were generated using the five feature subsets obtained using the hybrid FS approach. Next, the seven state-of-the-art supervised classification algorithms, namely SVM, KNN, NN, NB, DT, XGBoost, and LR, were trained and tested on the five subsets of the microarray data using ten-fold cross-validation to test the predictive potential of the selected subsets of gene biomarkers and thereby determine the most efficient breast cancer diagnostic model with higher accuracy, F1-Score, and AUC value. 

Finally, the supervised classification model’s predictive ability was estimated using a set of statistical matrices whose mean and standard deviation values are documented in the result section of the present article. Figure 1 shows a schematic representation of gene microarray data’s hybrid FS for primary BC tumor classification.

The remaining part of this research article is organized as follows: Section 3 includes the methodology of the research paper that has (1) the description of the datasets documented, (2) a detailed discussion of data preprocessing, (3) an explanation of the proposed hybrid feature selection strategy, (4) a description of the filter based FS algorithm and the metaheuristics techniques employed for developing the proposed hybrid FS strategy, (5) a description of the various machine learning classifiers used in the study, and (6) a description of the model performance evaluators. Section 4 discusses the results of the hybrid FS approach and the performance evaluation of the various classifier-based models. Section 5 discusses the results and compares the performance against available Hybrid FS-based ML models for earlier detection of BC patients. Finally, Section 6 involves the concluding remarks, future scope, and the limitations of the present study.

## 2. Literature Review

Many Hybrid FS methods involving heuristic algorithms have been developed in recent years to resolve the breast cancer classification problem. Here, we present and discuss the recently developed hybrid FS methods available in the literature.

As shown in Table 1, Shaban et al. [23] developed a new hybrid feature selection method (NHFSM) that takes advantage of the filter and wrapper FS methods. Firstly, in the preselection stage, the relevant nonredundant genes were selected using information gain (IG), a filter method. Next, the selected non-redundant features using IG were processed using a hybrid bat algorithm and particle swarm optimization (HBAPSO). The most optimal feature subset was then selected using the maximum fitness value. Lastly, a MATLAB tool was used to evaluate the efficacy of the proposed NHFSM method, which computed the value of the different performance measures such as accuracy, sensitivity/recall, F-measure, precision, and error rate. The NHFSM method enhances the accuracy of BC patients’ classification compared to the state-of-the-art feature selection approaches by obtaining an accuracy of 0.97. Tahmouresi et al. [24] proposed an FS method that merges gene rank and improved Binary Gravitational Search Algorithm (iBGSA), and the combination was named a pyramid Gravitational Search Algorithm (PGSA). The authors observed that the suggested FS method outperformed other wrapper techniques, having more than 70% of features reduced from the original number of features. Hamim et al. [25] suggested a hybrid method for gene biomarker selection from the gene microarray data.

The method combines a Fisher Score-based filter and the Ant Colony Optimization (ACO) algorithm. The Hybrid FS method was named the Hybrid fisher-Ant Colony Optimization (HFACO) algorithm. The combination of HFACO with the C5.0 classifier showed high classification accuracy. Ghozy et al. [26] proposed a hybrid FS framework using Information Gain (IG) as a filtering method and Genetic Algorithm (GA) as a wrapping method to decrease the dimension of the feature set. Finally, classification was performed using a Functional Link Neural Network (FLNN).

The experimental result showed that the authors could screen 49 gene biomarkers to classify breast cancer with an accuracy of 85.63%. AbdElNabi et al. [27] proposed an FS method that comprises IG and Grey Wolf Algorithm (GWO) to detect BC from microarray data by applying an SVM classifier. Tang et al. [28] developed a novel FS strategy to detect extremely tissue-specific biomarkers, namely miRNA and DNA methylation markers. They then used the random forest approach to construct a classifier that can efficiently predict the origin of breast tumors. Jain et al. [29] created a hybrid gene selection method employing a combination of Correlation-based FS (CFS) with improved-Binary Particle Swarm Optimization (iBPSO). The hybrid CFS-iBPSO method and NB classifier attained greater than 90% accuracy in the BC microarray dataset. Using the hybrid method, out of 24,481 genes in the data, only thirty-two differentially expressed genes from the BC dataset were screened, which equates to only 0.13% of the original number of genes in the BC dataset. Shukla et al. [30] proposed a framework for gene selection that combines Conditional Mutual Information Maximization (CMIM) and Adaptive GA (AGA). The suggested gene selection approach was tested on the BC microarray dataset. The results of the CMIMAGA method with the Extreme Learning Machine (ELM) classifier exhibited the maximum classification accuracy among other classifiers.

Lu et al. [31] proposed a gene selection method that combines Mutual Information Maximization (MIM) and adaptive GA (MIMAGA). The suggested approach successfully reduced the original 20,000 genes from the BC data to below three hundred with a mean accuracy of 80% in classifying the target variable. Mohapatra et al. [32] proposed a novel FS method for microarray data built on the Modified Cat Swarm Optimization (MCSO) algorithm. The results showed that MCSO-based Wavelet Kernel Ridge Regression (WKRR) outclassed other classifiers for the microarray dataset employed in their research. Finally, Shreem et al. [33] introduced a hybrid FS method using the Symmetrical Uncertainty (SU) with Harmony Search (HS) algorithm. The introduced method selected less than thirty genes for the detection of BC. Table 1 summarizes different gene selection techniques for predicting the earlier detection of BC using various microarray datasets.

## 3. Materials and Methods

### 3.1. Dataset and Preprocessing

The Breast_GSE22820 gene expression microarray data were used for hybrid FS, and supervised classification models were downloaded from the CuMiDa database [10]. The Breast_GSE22820 dataset is preprocessed data provided by CuMiDa, where data has been carefully and manually curated from sample quality, undesirable probes, background correction, and finally normalized to generate a more trustworthy resource of gene expression microarray data for computational cancer genomic studies. The Breast_GSE22820 comprised 33,580 gene expression profiles of 186 samples (10 normal breasts and 176 primary breast cancer patient samples) [48]. The target classes, namely the normal breast and primary breast cancer samples, were transformed into categorical string variables, i.e., Normal breast samples encoded as “0” and Primary Breast cancer samples encoded as “1”. Furthermore, the sample column variable was removed from the original Breast_GSE22820 dataset. Finally, the Quasi constant technique was applied to the transformed Breast_GSE22820 dataset variables to screen and filter similar variables using a variance threshold of 0.01. After applying the quasi constant, 2184 variables were filtered, and a subset of data comprising the remaining features and the class variable was created. The filtered Breast_GSE22820 dataset with 10 normal breasts and 176 primary breast cancer patient samples represents a class imbalance problem where one of the classes is under represented. One approach to address the imbalance of classes in the Breast_GSE22820 dataset is to oversample the minority class (normal breasts sample encoded as “0”). In our paper, we have used a popular data augmentation technique named the Synthetic Minority Oversampling Technique (SMOTE) [47,49] to artificially synthesize new sample values for the minority class (normal breasts sample) variable. The SMOTE parameter setting for performing oversampling of the minority class is as follows: sampling strategy = auto, random_state = None, *k*-neighbors = 5, and n_jobs = None. In the transformed Breast_GSE22820 dataset, upon oversampling, the number of samples of each class (encoded as 0 and 1) is balanced (1:176, 0:176). The balanced data is divided into 80 % training and 20 % testing data. Table 2 shows the distribution of the balanced gene expression dataset in training and testing data across the two classes (primary breast tumor and normal breast).

### 3.2. Proposed Hybrid FS Strategy

The proposed hybrid FS method involves an mRMR, a two-tailed unpaired *t*-test, and five state-of-the-art meta-heuristics algorithms to screen the most optimal and stable subset of features for the classification objective, depicted in Figure 2a. Furthermore, a detailed illustration of the framework is employed to screen the most optimal subsets of features to train and test various supervised classification models which classify the two classes of the Breast_GSE22820 gene expression microarray data with better accuracy, F1-Score, and AUC value, which is shown in Figure 2b.

In the proposed hybrid FS method, firstly, the multivariate filter-based mRMR technique using the F-statistic with a Correlation Quotient (FCQ) scoring scheme was used to screen the most relevant and nonredundant feature subset. Eventually, a subset of the twenty most relevant and nonredundant features was selected based on the FCQ score. A subset of the original data, including the twenty-one features (20 independent features + 1 dependent variable), was then generated and used for further analysis. Secondly, a two-tailed unpaired *t*-test was performed to compare the mean of the distribution among two classes (primary breast cancer samples and normal breast samples) of the twenty most relevant and nonredundant features screened using mRMR at a 5% significance level. Next, a subset of data was generated, including the class variable and features with a significant mean difference between the two classes (groups) at a 5 % significance level. Thirdly, five state-of-the-art meta-heuristic algorithms were used to screen the most optimal feature subsets from a subset of features whose average distribution values are significantly different between the two classes (groups). Finally, the five most optimal feature subgroups obtained using the five meta-heuristic algorithms were trained and tested using stratified ten-fold cross-validation on various supervised classification algorithms to develop supervised ML models that can classify the primary breast cancer samples from the normal breast cancer sample with higher accuracy, F1-Score, and AUC value.

#### 3.2.1. FS Based on mRMR (FCQ)

The mRMR [50] is a minimal-optimal multivariate filter-based FS method that tends to screen a subset with a minimal number of most relevant (high relevance with the response variable) and nonredundant features (minimum correlation between the selected features). In the present study, the Breast_GSE22820 microarray data and the genes (features) expression data are continuous, and response variable labels are binary category variables (0 and 1). Therefore, the relevance and redundancy information of mRMR among selected gene biomarkers and class labels (0 and 1) in the present study is measured by F-statistic and Pearson’s coefficients, respectively. The working of the mRMR algorithm is represented using the following equations:

For continuous attributes (individual gene expression data), the F-statistic between the genes and the target variable *h* can be chosen as the maximum relevance score. The *F*-test value of gene (feature) *g_i_* in *K* classes represented by *h* is written as Equation (1):(1)$$F(g_{i},\;h)=\underset{k}{[}\sum n_{k}({\bar{g}}_{k}-\bar{g})/(K-1)]/s^{2}$$

Here $\bar{g}$ is the average value of a gene ($g_{i}$) in each tissue sample, ${\bar{g}}_{k}$ is the average value of $g_{i}$ within the *k*th class, *K* is the number of classes, and $\sigma^{2}=\underset{}{[}\sum_{k}{(n}_{k}-1)\sigma^{2}{}_{k}](n-K)$ is the collective variance (here $\sigma_{k}$ and ($n_{k}$) and are the variance and size of the *k*th class). Using the relation $F=t_{2}$, the *F*-test will reduce to the *t*-test for the two-class classification problem. Consequently, for the gene biomarker set *S*, the maximum relevance can be represented as:(2)$$max\;V_{F},\;V_{F}=\frac{1}{\left|S\right|}\sum_{i\in S}F(i,\;h)$$

The present study specifies the minimum redundancy condition between the input variables using the Pearson correlation coefficient ${c(g}_{i}{,g}_{j})=c(i,j)$. Here both high negative and high positive correlations’ mean redundancy is considered. Therefore, the absolute value of these correlations has been taken, and the given condition (minimum relevance between the input variables) has been represented below in Equation (3).
(3)$$min\;W_{c},\;W_{c}=\frac{1}{{\left|S\right|}^{2}}\sum_{i,j}\left|c(i,j)\right|,$$

Next, the following features are screened using the mRMR optimization criterion function involving FCQ. Finally, in Equation (4), we combine *F*-test with Correlation using Quotient, which is written as:(4)$$\underset{i\in \Omega_{S}}{max}\text{\{}F(i,h)/[\frac{1}{\left|S\right|}\sum_{j\in S}\left|c(i,j)|\right]\text{\}}$$

The precise solution to the MRMR requirements needs exploration (*N* is the number of genes in the whole gene set, Ω). Normally, a near-optimal solution is enough. Therefore, a simple linear incremental search (heuristic) algorithm is required to solve the MRMR optimization function and choose the following features assuming we have already selected m features. 

In the present study, we selected the top 20 features (*S* = 20) as the size of the minimal-optimal feature subset. Subsequently, a data subset was generated comprised of twenty genes (features) from the mRMR algorithm, which was further analyzed using a Student’s *t*-test to select features whose mean significantly differed between the target class sample populations (primary breast cancer samples and normal breast samples).

#### 3.2.2. FS using a Two-Tailed Unpaired *t*-Test

A two-tailed unpaired *t*-test [51] at a 5% significance level was performed on the subset data with twenty-one features (twenty features from mRMR and one class variable). The two-tailed unpaired *t*-test was executed to screen features that demonstrate significant differences between the mean values of the feature across the two classes’ populations.

#### 3.2.3. FS Using Meta-Heuristic Algorithms

In the present study, five state-of-the-art meta-heuristic algorithms, namely, Equilibrium Optimizer (EO), Binary Bat Algorithm (BBA), Cuckoo Search Algorithm (CSA), Red Deer Algorithm (RDA), Genetic Algorithm (GA), were used to obtain the most optimal feature subset from the subset of a dataset comprising the significant features set attained using a two-tailed unpaired *t*-test. The five state-of-the-art meta-heuristic algorithms, namely EO, BBA, CSA, RDA, and GA, are discussed below.

Binary Bat Algorithm

The bat moves in a search space toward continuous-valued locations. However, in the selection, the bat travels across the corners of a hypercube since the search space is modeled as an 𝑛- dimensional Boolean framework [52]. Meanwhile, the bat’s position is represented by binary vectors since the optimization problem is to choose or not choose a particular feature. Thus, the Bat Algorithm restricts the new bat’s location to binary values by employing a sigmoid function [53], as shown below:(5)$$S(v_{i}^{j})=\frac{1}{1+e^{{-v}_{i}^{j}}}$$

Suppose for each bat bi an initial position 𝑥𝑖, velocity 𝑣𝑖, and frequency 𝑓𝑖 are initialized. Then, for an individual iteration, say 𝑡, where 𝑇 is the maximum number of iterations, the movement of the virtual bats is represented by revising their position and velocity by employing Equations (6)–(8), as shown below:(6)$$f_{i}=f_{min}+(f_{min}-f_{max})\beta ,$$
(7)$$v_{i}^{j}(t)=v_{i}^{j}(t-1)+[{\hat{x}}^{j}-x_{i}{}^{j}(t-1)]f_{i},$$
(8)$$x_{i}{}^{j}(t)=x_{i}{}^{j}(t-1)+v_{i}^{j}(t),$$

Here 𝛽 signifies an arbitrarily created number between 0 and 1. It signifies the value of the decision attribute denoted by 𝑗 for a given bat 𝑖 at an iteration 𝑡. The outcome of $f_{i}$, as shown in Equation (1), is utilized to regulate the range and pace of the bat’s movement. The variable ${\hat{x}}^{j}$ depicts the present global best solution (global minima) for the decision attribute 𝑗, which is achieved by comparing all the positions (solution) presented by the 𝑚 bats. The binary Bat Algorithm limits the new bat’s location to a binary value by using a sigmoid function, as shown in Equation (9). Thus, Equation (8) can be substituted by:(9)$$x_{i}{}^{j}(t)=\left\{ \begin{array}{l} 1\;if\;S(v_{i}^{j})>\sigma ,\\ 0\;otherwise \end{array}\right.$$

Here σ ~ U(0,1). Thus, Equation (9) can offer only binary values for a given bat’s coordinates within the Boolean framework, which stands for choosing or not choosing the features.

2.Genetic Algorithm

The GA is a heuristic search-based optimization algorithm inspired by Charles Darwin’s theory of evolution and natural selection. The GA is used to find the optimal or near-optimal solution to NP problems that are difficult to solve [54,55,56]. In GA, we have a set of solutions for a given problem. Each candidate solution (variables) has a fitness value based on the corresponding objective function. The fitter individuals (solutions) are more likely to mate and produce more “fit” individuals (solutions). The GA concept aligns with the Darwinian theory of “Survival of the Fittest.” These screened solutions (parents) then undergo crossover (recombination) and mutation, generating new solutions with better fitness values. This process repeats over various generations (iterations), and eventually, the selection process is terminated once a generation with the fittest (optimal) solution is found. The complete procedure GA is illustrated by a flow diagram, as shown in Figure 3.

3.Equilibrium Optimizer Algorithm

The EO algorithm is a new meta-heuristic algorithm developed in 2019 for solving single-objective optimization problems [57]. EO uses the best-so-far solutions (equilibrium candidates) to update each particle (solutions) to reach an optimal solution (equilibrium state) finally. The EO algorithm uses the well-defined “generation rate,” which enables the EO to have strong exploration and exploitation abilities and local optimum avoidance. The structure of EO is simple and easy to implement. Moreover, the EO algorithm is computationally effective, as its complexity is of polynomial order:(10)O(EO)=O(1+nd+tcn+tn+tnd)≅ O(tnd+tcn)

Here, *O* = Big − O notation is used as common terminology, *t* = a number of iterations, *n* = a number of particles, *d* = number of dimensions, and *c* = cost of function evaluation.

4.Cuckoo Search Algorithm

The cuckoo search is a metaheuristic optimization algorithm developed for solving optimization problems [58,59]. It is a nature-inspired algorithm centered on the brood parasitism of certain cuckoo species using levy flights-random-walks to lay their eggs in the nests of host birds of another species. Moreover, the cuckoo search is a specific case of the familiar (μ + λ)-evolution approach [60]. The CS algorithm employs the following depictions:

In a straightforward case, we assume that each nest holds one egg. An individual egg in a nest signifies a solution, and an additional cuckoo egg represents another solution. The CS algorithm aims to replace a not-so-good solution (host egg) in the host nest with a better solution (cuckoo egg). The algorithm can be further expanded to a more complex scenario where each nest consists of many eggs representing multiple solution sets.

Additionally, the CS algorithm is built on three central rules:

An individual cuckoo lays only one egg at a time and leaves its egg in an arbitrarily selected nest of a host species;The finest nests with superior-quality eggs will be passed down to the subsequent generation;The number of accessible hosts and their corresponding nests are set, and the probability of an egg laid by a cuckoo to be discovered by the host bird is given as follows:(11)$$p_{a\;}\in \;\left(0,1\right)$$

In the present scenario, the host bird can put the cuckoo egg away or desert the nest, or the host can construct a new nest.

A significant benefit of the CS algorithm is its simplicity. Compared with other agents- or population-based meta-heuristic algorithms, namely harmony search and particle swarm optimization, there is an additional single parameter *p_α_* more in the CS algorithm along with the parameter, which is the population size *n*. Thus, the CS algorithm is significantly simpler in implementation.

5.Red deer Algorithm

The Red Deer algorithm (RDA) is a population-based meta-heuristic algorithm [61]. The RD algorithm is nature-inspired. The RD algorithm combines the competence and strength of heuristic search techniques and the survival of the fittest concept of the evolutionary algorithms [62]. The main inspiration of the meta-heuristic RD algorithm originates from an uncommon mating manner in a breading season by the Scottish red deer. Like the other population-based meta-heuristics, the RDA starts with an initial population (solutions) randomly distributed, and each solution is called Red Deer. Here, each red deer is considered to have a subset of features. The fitness of the red deer (individual feature subset) is calculated, sorted, and the best fittest feature subset (red deer) is selected as stags (N_m_). The remaining are classified as hinds (N_f_) using Equation (12).
(12)$$f_{i}=\left\{ \begin{array}{l} 1\;if\;r>0.5\\ 0\;otherwise \end{array}\right.$$

Where $f_{i}$ represents the discrete format of the fitness vector. *r* is a continuous random variable between 0 and 1

The stages are employed to exploit feature space, and the hinds are utilized to explore the feature space. Exploitation by stags is confirmed by assessing the nearby stags made using Equation (13) given below.
(13)$$Stag_{new}=\left\{ \begin{array}{l} Stag_{old\;}+r1*((UB-LB)*r2)+LB\;if\;r3\geq 0.5\\ Stag_{old\;}-r1*((UB-LB)*r2)+LB\;if\;r3\;<0.5 \end{array}\right.$$

The Upper Boundary (UB) and the Lower Boundary (LB) of the search space represent the highest number of features that can be selected and the lowest number of features that must be selected, respectively. The variables *r*1, *r*2, and *r*3 signify continuous random values ranging from 0 to 1. The current fittest solution (Stag) is designated as $Stag_{old}$, and the adjacent stag is designated $Stag_{new}$. If the fitness value of the $Stag_{new}$ is greater than the fitness value of the, then the $Stag_{new}$ will replace the $Stag_{old}$ as the fittest solution (best feature subset). Finally, the best solution (deer) amongst the stag is chosen as the commander. The commander then breeds 80% of the female (hind) in the feature space (population) to generate offspring. Equation (14) shows the commander’s breeding with the hind.
(14)$$RD_{new}=\frac{\left(Com+Hind\right)}{2}+(UB-LB)*c$$

$RD_{new}$ is the new Red deer (new solution or feature subset). The commander and hinds in Equation (5) are designated by Com, and Hind, respectively. The variable “*c*” signifies a continuous random variable with its value ranging from 0 to 1. The breeding process results in an acceptable level of exploration or diversification of all possible solutions. Moreover, the present commander will fight with randomly chosen stags in the feature space (population). The stag who defeats the current commander will become the next commander (optimal solution). Next, the fitness value of all the existing stags is calculated again. The process endures until the commander deer obtains the maximum fitness value or the greatest number of deer is generated. The pseudocode of the feature selection process used in the RD algorithm is shown below in Figure 4:

### 3.3. Feature Selection Methods Parameter Setting

All FS experiments were performed on a laptop with an Intel Core i7 processor with 2.4 GHz and 8 GB of RAM. Table 3 shows the configuration of the parameters settings of the FS algorithms based on filter and nature-inspired metaheuristics algorithms.

### 3.4. Partitioning of the Microarray Datasets Obtained Using Hybrid FS Techniques

Each of the five subsets of the gene microarray dataset generated from the five optimal feature subsets obtained after the hybrid FS techniques was partitioned into 80% training and 20% independent testing datasets. Each of the 80% training datasets consists of 282 samples, of which 141 samples belong to primary breast cancer, and the other 141 samples belong to normal breast samples, as shown in Table 2. Stratified ten-fold-training-cum-cross-validation on the 80% training sets was used to train seven supervised classification models: DT, RF, KNN, SVM, XGBoost, Gaussian-NB, and LR. To avoid prediction bias, it is mandatory to test the trained models on independent test data unseen by classification models during the model training phase.

Subsequently, stratified ten-fold cross-validation on the 20% testing datasets was performed to assess the classifying proficiency of each ML-based trained model. All five 20% testing data derived from the five heuristics FS method consist of 35 samples belonging to the primary breast cancer samples and the other 35 samples belonging to the normal breast samples, as shown in Table 2. The mean and standard deviation of each performance evaluator (accuracy, F1-Score, and ROC-AUC) were recorded and compared to select the best set of features and the corresponding feature subset-based model that can classify primary breast cancer samples from the normal breast samples with higher accuracy, F1-Score, and AUC value. Each testing dataset consisted of 70 samples, while 35 belonged to each class label: primary breast cancer and normal breast.

### 3.5. Training Classification Algorithms

The seven typically used supervised classification algorithms used for the training of the seven different classification models are discussed below:

#### 3.5.1. Logistic Regression

Logistic regression is a supervised learning classification algorithm. The LR classifies discrete target variables based on a sigmoid function, where the independent variable’s values lie between +∞ to −∞, and the output is a probabilistic value of a target variable that ranges between 0 to 1 [63,64]. The mathematical equation of the LR is shown below:(15)$$y=\frac{e^{\left(b_{0}+b_{1}X\right)}}{1+e^{\left(b_{0}+b_{X}\right)}}$$
$$\begin{array}{l} Here,\;x=input\;value;\;y=predicted\;output;\;b_{0}=bias\;or\;intercept\;term;\;b_{1}=coefficient\;\\ for\;input\;variable\;\left(x\right) \end{array}$$

#### 3.5.2. XGBoost

Extreme Gradient Boosting, abbreviated as XGBoost, is an accurate implementation of distributed gradient-boosted decision tree (GBDT). The XGBoost algorithm implements a parallel boosting tree and is the most popular method for solving classification, ranking, and regression problems [65]. XGBoost uses an ensemble of K-classification DTs. Each of the DTs has KiE|i ∈1…K nodes. The final classification score is the sum of the prediction score of each DT and is shown below using the following equation:(16)$$\hat{y_{i}}=\phi (X_{i})=\sum_{k=1}^{K}f_{k}(X_{i}),\;fk\;\in \;F$$

Here, $X_{i}$ refer to instances of the training dataset and $y_{i}$ represent the target-dependent variable labels, $f_{k}$ denotes the leaf score for the *k*th DT, and *F* represents the set of each *K* score for the whole classification DTs. The result of the model is improved using regularization and is shown below using the following equation:(17)$$L\left(\varphi \right)\;=\sum_{i}l\left({\hat{y}}_{i},\;y_{i}\right)\;+\sum_{k}\Omega {\;(f}_{k})$$

Here, the first part of the equation measures the difference between the target and the predicted n ${\hat{y}}_{i}$ using the differentiable loss function represented. The second part of the equation “Ω” prevents over-fitting by penalizing the complexity of the model and is depicted using the following equation:(18)$$\Omega (f)=\gamma T\;+\frac{1}{2}\lambda {\left\|w\right\|}^{2}$$

Here, the quantity of leaves in the tree is denoted by *T*, the weight of each leaf is represented by *w*, and the constants regulating the degree of regularization are represented by γ, and λ, respectively.

#### 3.5.3. Gaussian Naïve Bayes

The GNB is a modified version of NB. Gaussian distributions [66,67] are a typical method to describe the likelihoods of the continuous variable conditioned on the target variable in the NB classification. Therefore, each independent variable in the GNB algorithm is specified by a gaussian Probability Density Function (PDF) and is represented by the following equation:(19)$$X_{i}\;∼\;N\left(\mu ,\;\sigma^{2}\right)$$

The Gaussian PDF takes the form of a bell and can be represented by the following equation:(20)$$N\left(\mu ,\;\sigma 2\right)\left(x\right)\;=\frac{1}{\sqrt{2\Pi \sigma^{2}}}e^{-\frac{{\left(x-\mu \right)}^{2}}{2\sigma^{2}}}$$

Here, *μ* represents the mean, and *σ*2 is the variance. In GNB, we must define a normal or Gaussian distribution for each continuous variable. The parameters of such normal distributions can be obtained using the following formulae:(21)$$\mu X_{i}|C=c\;=\frac{1}{N_{c}}\sum_{i=1}^{N_{c}}x_{i}$$
(22)$$\mu X_{i}|C=c\;=\frac{1}{N_{c}}\sum_{i=1}^{N_{c}}x_{i}{}^{2}-\mu^{2}$$

Here, *N_c_* is the number of instances where *C* = *c* and *N* represents the total number of training instances. Calculating *P* (*C* = *c*) for the target variable is performed by applying relative frequencies by using the following equation shown below:(23)$$P(C=c)=\frac{N_{c}}{N}$$

#### 3.5.4. Decision Trees

DT is a nonparametric supervised machine learning algorithm that generates a classification model that classifies the target variable by evaluating a tree of true/false or if-then-else independent variable questions. The DT algorithm estimates the least number of variable questions needed to reach the decision point (leaf node). The DT algorithm can be used as a regressor to predict a continuous numeric value of a target variable or as a classifier to classify a categorical target variable [68]. Our study uses the DT algorithm to classify the normal breast sample encoded as 0 from the primary breast cancer encoded as 1. In our study, we have employed the Gini Impurity [69], a classification metric, to measure the quality of a split and how the internal and leaf nodes will be generated using the DT classification algorithm. The formula for calculating the Gini index is shown below:(24)$$Gini=1-\sum_{i=1}^{n}{\left(p_{i}\right)}^{2}$$

*p_i_* is the probability of an instance being classified by the DT algorithm to a particular class label.

#### 3.5.5. K-Nearest Neighbor

The classic KNN algorithm is a nonparametric supervised classification technique applied to solve classification problems [70]. The algorithm has a variable parameter known as “*k*”, which tells us about the number of “nearest neighbors” or data points from the training data. The nearest neighbors are found based on the nearest distances from the query data point. Euclidean Distance (ED) techniques calculate the distance between the two data points. After finding the *k* closest data points or neighbors, the algorithm executes a majority voting rule to find the target variable with maximum appearances (hits). The class label with maximum appearances is declared to be the final classification class label for the query. The equation for the calculation of the distance function (Euclidean distance) is shown below:
(25)$$Euclidean\sqrt{\sum_{i=1}^{k}{\left(x_{i}-y_{i}\right)}^{2}}$$

The ED is calculated as the square root of the sum of the squared differences between a new data point (*x*) and an existing data point (*y*).

#### 3.5.6. Support Vector Machine

The SVM is another nonparametric machine learning algorithm that can categorize an unlabeled data point by obtaining a suitable hyperplane in an n-dimensional feature space. However, the SVM output is not nonlinearly separable. Thus, when employing SVM in data analysis, selecting appropriate parameters and kernels is vital to prevail over such problems [71]. In addition, the SVM algorithm helps classify class labels for small-size data where the number of instances is comparatively lesser than the number of independent variables in the training dataset.

#### 3.5.7. Random Forest

An ensemble learning RF algorithm is widely employed for classification and regression-supervised learning tasks. The Random Forest classification technique uses an ensemble of DTs to predict a sample (query) class label. The training set of the original data is divided into smaller groups, and each subgroup builds an individual decision tree in an RF. Each DT generates an outcome [72]. The class label with the maximum votes from each DT turns into the final chosen class label by the RF algorithm. The RF algorithm utilizes bagging and randomness attributes during the creation of each DT for developing a noncorrelated forest of DTs. Here, the classification by the forest is significantly more accurate than any single DT. Additionally, the RF algorithm demonstrates better prediction accuracy for small-size data.

Suppose, for a nominal splitting feature, $X_{i}$ and $X_{i}$ denoted possible levels as to $L_{j}$. Therefore, the Gini index for the feature $X_{i}$ is computed using the following equation:(26)$$G(X_{i})=\sum_{j=1}^{j}Pr(X_{i}=L_{j})(1-Pr(X_{i}=L_{j}))=1-\sum_{j=1}^{j}Pr{\left(X_{i}=L_{j}\right)}^{2}$$

### 3.6. Machine Learning Algorithms Parameter Setting

The final parameter settings used to construct each ML model are shown in Table 4.

### 3.7. Classification Model Performance Metrics

The classifying capability of the supervised classification models was evaluated using the following classification model performance metrics:

#### 3.7.1. Accuracy and Confusion Matrix

Accuracy plays a vital role in assessing classification models in a scenario where the samples of each class are equal in number. In our study, the test dataset is balanced. Therefore, accuracy offers a better insight into the model’s capability to classify the target class. The accuracy value ranges from 0 to 1, where the model with an accuracy value of 1.0 is considered the best-performing model. On the contrary, the model with a value ‘of 0.0’ is considered the worst-performing model. Mathematically, the accuracy is determined using the following formula:(27)$$Accuracy=\frac{TP+TN}{TP+TN+FP+FN}$$

In Equation (23), true positive, true negative, false negative, and false positive are abbreviated as *TP*, *TN*, *FN*, and *FP*, respectively. Moreover, a confusion matrix defines the classification model’s performance on testing data for which the actual values are well-known. The related terminologies from the confusion matrix are defined as follows:

Positive class (in the present study, the normal breast).Negative class (in the present study, the primary breast cancer).True positive is a classification outcome where the classification model correctly classifies an actual positive instance (primary breast cancer) as a positive class sample.A true negative is a classification outcome where the classification model correctly classifies an actual negative instance (normal breast) as a negative class sample.A false positive is a classification outcome where the classification model incorrectly predicts a positive class sample as a negative class.A false negative is a classification outcome in which the model incorrectly predicts an actual positive class instance as a negative one.

Furthermore, a confusion matrix explains the classification models’ error types (type I and II) [73]. The present study aims to develop a primary breast cancer classification model that categorizes primary breast cancer (positive class) from normal breast samples. Therefore, other performance matrices, namely the F1-Score and AUC values, could provide greater insight into the classification model’s capability to classify a target class than accuracy alone [74].

#### 3.7.2. F1-Score as a Model Performance Evaluator

The F1-Score is defined as the harmonic mean of precision and recall and is calculated using the following mathematical formula:(28)$$F1\;score=\frac{2\times \mathrm{Precision}\times \mathrm{Recall}}{\mathrm{Precision}+\;\mathrm{Recall}}$$

#### 3.7.3. The Area under the Receiver Operating Characteristic Curve (ROC-AUC)

The Receiver Operator Characteristic (ROC) curve is a metric typically used to evaluate a binary classification problem. The ROC is a probability curve that plots a two-dimensional graph between the True Positive Rate (*TPR*) and the False Positive Rate (*FPR*) at decision threshold values (ranging from 0 to 1) and essentially separates the ‘noise’ from the ‘signal’. The *TPR* and *FPR* are calculated using Equations (25) and (26), respectively.
(29)$$TPR=\frac{TP}{TP+FN}$$
and
(30)$$FPR=\frac{FP}{FP+TN}$$

The AUC is an estimation of the capability of a classification model to distinguish between the two classes and is used to summarize the ROC curve. The AUC value determines the complete two-dimensional area under the ROC curve, starting from the coordinates (0, 0) and finishing at (1, 1), thereby allowing an accurate measurement of the model’s ability to differentiate the two class labels in the gene microarray dataset. The higher the AUC value, the better the implementation of the classification model to distinguish the positive and negative classes in a two-class classification problem. For example, a classification model with an AUC value of 1.0 can accurately differentiate between the two-class labels in a dataset. If, however, the AUC value is zero, then the classifier predicts all actual positive samples as negative and all true negative samples as positive.

### 3.8. Histogram Frequency Curve Plot

The histogram frequency curve plot describes the frequency distribution of continuous attributes belonging to the corresponding two class labels. In our study, the plot represents the frequency distribution of genes (continuous variables) between the two class labels population in the Breast_GSE22820 dataset. Additionally, the histogram frequency curve represents the difference in the population mean of the continuous attributes (genes) between the two class labels (primary breast cancer and normal breast samples).

### 3.9. One-Tailed Unpaired t-Test

We performed a one-tailed unpaired *t*-test [51] to evaluate the performance of our XGBoost-based classification model built using a subset of gene biomarkers obtained using a novel framework of the Hybrid FS method with the recently published machine learning models built using different gene biomarkers for an earlier prediction BC.

### 3.10. Web Application

Render is a cloud-based platform used to host our XGBoost-based classification application. The output of our application is a probabilistic score.

## 4. Results

### 4.1. Screening of Novel Set of Potential Gene Biomarkers

Despite the recent development in imaging technologies, the earlier detection of BC remains a challenge for 21st-century researchers. In the current setting, screening potential genes involved in the pathophysiology of BC will provide researchers and clinicians with a precise method to detect BC in the earlier stages of its existence in the human system. In this context, screening potential gene biomarkers using various FS techniques from the gene expression microarray data of BC patients has been proven helpful in recent times in understanding the molecular aspects of BC. Therefore, in this regard, screening potential gene biomarkers for accurately classifying primary breast cancer patients will enable less but more reliable testing, resulting in faster and more effective diagnostics and prognostic procedures in managing BC patients. In the present context, in our study, a hybrid FS technique was executed to screen potential gene biomarkers to classify primary breast cancer samples from the gene microarray data. A novel hybrid FS framework of a multivariate filter-based method, a Student’s *t*-test, and five state-of-the-art nature-inspired meta-heuristic methods, namely EO, BBA, CSA, RDA, and GA, was employed to screen a potential set of gene biomarkers for the earlier detection of primary breast cancer.

#### 4.1.1. The mRMR Feature Importance

The mRMR method selects a subset of features with minimum redundancy and maximum relevance with the target variables. The FCQ scoring variant of mRMR was used to calculate the correlation between continuous variables (genes) and the relevance of the variables to the target variable of the Breast_GSE22820 gene expression dataset. The top twenty features obtained using the FCQ scoring variant of the mRMR have been tabulated in Table 5.

#### 4.1.2. FS Based on Two-Tailed Unpaired *t*-Test

The two-tailed unpaired *t*-test at a 5% significance level was executed to screen features obtained from the mRMR algorithm, demonstrating a significant difference between the mean value of each twenty features between the two classes’ sample populations. The results of the two-tailed unpaired *t*-test of the mRMR selected the top twenty gene biomarkers evaluated at a *p*-value < 0.05 and is shown in Table 6.

The Gene Biomarkers (features) whose frequency distribution mean was significantly different (at a 5% significance level) between the two classes of labels that were selected are shown in Appendix A. Therefore, a subset of five features, namely NM_152426, BC016934, NM_138957, NM_001008493, and NM_006579, with a significant mean difference between the class population verified using unpaired *t*-tests, were selected and further screened using five state-of-the-art metaheuristic algorithms to obtain an optimal subset of features capable of discriminating the target variables with higher performance.

#### 4.1.3. Screening of Optimal Feature Subsets Using Metaheuristic Algorithms

The present study comprehensively analyzes the five nature-inspired global optimization algorithms for optimal FS. The present study implements five state-of-the-art optimization algorithms to find the five optimal feature subsets out of the entire search space (i.e., six features). The EO metaheuristic method yielded an optimal subset with three features: ‘NM_138957’, ‘NM_152426’, and ‘NM_001008493’. The BBA-based metaheuristic identified an optimal subset of two features, namely ‘NM_152426’ and ‘NM_138957’. Similarly, using the CSA algorithm, we obtained an optimal subset of two features, such as BC016934 and ‘NM_006579’.

Using the RDA metaheuristic method, we obtained an optimal feature subset with one feature, namely BC016934. Lastly, an optimal subset with one feature, such as NM_152426, was obtained using the GA algorithm. The five optimal subsets screened using the five global optimization algorithms are listed in Table 7. 

### 4.2. Screening of the Best Classification Model

Each of the five optimal subsets of feature(s) obtained using five metaheuristic algorithms were trained and tested on seven supervised classification algorithms, namely DT, NB, SVM, XGBoost, LR, KNN, and RF, using the five data subsets prepared using the five optimal subsets of features obtained from the post-metaheuristic-based FS. In addition, stratified five-fold cross-validation was applied to train and test classification models in each training and independent test dataset. The present study aims to screen the best-performing classification model in distinguishing the target variable labels built using the most optimal subset of the gene biomarker to facilitate the earlier detection of BC patients. Figure 5a–c represents the accuracy, F1-Score, and AUC values of the seven classification models developed using the features obtained by applying the five different metaheuristic algorithms.

We can observe from the comparative performance evaluation of the seven classification models that the XGBoost-based model developed using the optimal feature subset obtained from the EO algorithm is the best classification model with accuracy (0.976 ± 0.027), F1-Score (0.974 ± 0.030), and AUC (0.987 ± 0.025) shown in Figure 5a–c and tabulated in Table 8, respectively. In addition, a confusion matrix summarizing the performance of the XGBoost-based classification model evaluated on independent test data is depicted in Figure 6, where primary breast cancer is a positive class, and the normal breast sample is considered the negative class.

A summary of the baseline model performance of the eight models built using all of the features in the COVID-19 dataset is listed in Table 8. The XGBoost-based classification model performance estimated using matrices, namely, accuracy, F1 score, and AUC, was better than the baseline models built using a subset of five features obtained post unpaired two-tailed *t*-test and base classifiers, namely SVM, NB, DT, LR, KNN, RF, and XGBoost, as represented in Figure 5a–c and Table 8, respectively. Therefore, as per the results, it can be deduced that the subset of gene biomarkers, namely, ‘NM_138957’, ‘NM_152426’, and ‘NM_001008493’ discovered is the best-performing combination of features to accurately classify the primary breast tumor from the normal breast samples. Furthermore, the XGBoost-based model built using the optimal feature subset attained using the EO algorithm was identified as the best-performing model, thereby facilitating an earlier detection of BC by accurately classifying the primary breast tumor from normal breast samples.

Moreover, using a histogram plot, the mean difference in the frequency distribution of the best-performing gene biomarkers (screened using the EO algorithm) between the two classes (normal breast and primary breast tumor samples) is represented in Figure 7a–c. In addition, the mean distribution of the selected gene biomarkers in the patient group’s two population classes, namely the primary breast tumor and the normal breast samples, is statistically significant at *p* < 0.001, as shown in Table 3. Therefore, we can propose that the selected gene biomarkers can be used as prognostic gene biomarkers to enable the earlier detection of BC.

### 4.3. Comparative Performance of Our Model with Other Relevant Models

Our proposed XGBoost-based classification model performance was compared with the performances of the classification models built using different hybrid feature selection approaches involving filter, wrapper, and other FS methods, as tabulated in Table 6. The relative accuracy and AUC-based performance evaluation of our proposed model and the DNN-based models are tabulated in Table 6. We observe that the gene biomarkers selected using our proposed hybrid FS approach, which involves mRMR, a minimal-optimal filter-based FS algorithm, an unpaired *t*-test, a statistical method, and metaheuristics algorithms achieved the highest accuracy (0.976 ± 0.027) and AUC value (0.987 ± 0.025) as compared to classification models built using other gene biomarker-based classification models, as shown in Table 9.

### 4.4. Web Application

The XGBoost-based classification model was implemented as a web application and hosted at https://appbcgene.onrender.com/, accessed on 1 December 2022. In the real-time online primary breast tumor detector, the users can classify primary breast tumor samples from normal breast samples using the gene expression values of the genes, namely MAPK-1, APOBEC3B, and ENAH actin regulator (ENAH). Therefore, clinicians can use our online breast tumor predictor to detect BC in its earlier stages.

## 5. Discussion

Screening robust gene biomarkers from gene microarray data for earlier detection of BC has long been a challenging problem due to the low sample size and multi-dimensionality properties of gene microarray data. We used hybrid learning in our FS framework to screen robust gene biomarkers. We systematically evaluated five hybrid FS approaches by introducing a sequential FS pipeline comprising a filter-based method (mRMR) followed by a statistical method (a two-tailed unpaired *t*-test). Finally, the subset of statistically significant features (gene) obtained using the two-tailed unpaired *t*-test was processed) using the five state-of-the-art metaheuristic algorithms (EO, BBA, CSA, RDA, and GA). Various supervised classification algorithms, namely SVM, NB, DT, LR, KNN, RF, and XGBoost, were used to screen the final most optimal feature subset of predictors for classifying primary breast tumors from normal breast gene expression microarray data. The current approach is a comprehensive analysis of a hybrid approach involving a sequential application of a filter-based method, statistical, and metaheuristics for gene FS for earlier detection of primary breast tumors. Compared to other hybrid FS methods and ML models, as tabulated in Table 1, our proposed hybrid-based FS approach and XGBoost classification model resulted in the most significant enhancement in overall accuracy and AUC value.

In this study, the hybrid-based FS pipeline screened an optimal subset that includes three gene biomarkers, namely MAPK1, APOBEC3B, and ENAH, for earlier detection of primary breast tumors. In recent years, clinical studies have shown that these screened gene biomarkers, namely MAPK 1 [75,76,77,78,79,80], APOBEC3B [81,82,83,84], and ENAH [85,86,87,88], are known to be related to the pathophysiology of BC. Therefore, in addition to an unpaired *t*-test that characteristically focuses on each gene’s differential expression between two classes of samples, we also focused on the differences in their ontology. We performed an ontology analysis [89,90] of the three genes screened using our hybrid method to explore the pathways of the three genes and their role in breast cancer pathogenesis. The ontology studies show that the MAPK1 gene plays a vital role in MAPK, an intracellular signaling pathway. Among various intracellular signaling pathways, the MAPK pathway is more important in promoting cell proliferation, cell differentiation, angiogenesis, cell survival, and tumor metastasis than other pathways in the tumorigenesis of BC. Four major MAP kinase signal transduction pathways exist in human cells. However, the ones concerning ERK-1 and ERK-2 of the ERK/MAPK signaling pathway are most relevant in tumor formation in BC [75]. In addition, recent studies have demonstrated elevated ERK expression in an increased proportion of cells in human tumors, such as breast cancer [76,77].

Moreover, the MAPK signaling pathway plays a pivotal role in the pathogenesis of TNBC [78]. MAPKs also participate significantly in the expression of PR, HER-2, and ER and are strongly associated with the infiltration and metastasis of TNBC [79]. Furthermore, in TNBC, the MAPK and EGFR act synergistically to promote TNBC progression, and their higher expression levels in TNBC patients (compared with paratumor tissues) are strongly correlated with lymph node advanced clinical stage, metastasis, recurrence metastasis, and poor prognosis [80].

DNA cytosine deaminase APOBEC3B was recognized recently as a cause of DNA mutagenesis and DNA damage in head/neck, breast, lung, cervix, bladder, and ovary cancer [81]. The APOBEC3B enzyme is usually an effector protein in the primary immune reaction to viruses. However, upregulation of APOBEC3B in the head/neck, breast, lung, cervix, bladder, and ovary cancer results in raised levels of Cytosine-to-Uracil deamination cases. These events mainly manifest as transitions of Cytosine-to-Thymine and transversions of Cytosine-to-Guanine within a specific DNA trinucleotide pattern (favorably 5′-TCG and 5′-TCA). The APOBEC3B catalyzed genomic deamination events leading to the transition of Cytosine-to-Uracil within the trinucleotide pattern (5′-TCG and 5′-TCA) also precede kataegis (cytosine mutation groups), leading to visible possible chromosomal abnormalities, namely translocations.

Furthermore, clinical investigations show that the higher expression of APOBEC3B is associated with a poorer survival rate for estrogen receptor-positive breast cancer patients, which includes brief periods of disease-free survival and, in particular, survival post-surgery [82]. Therefore, APOBEC3B can be considered a potential diagnostic and prognostic biomarker in BC. In addition, APOBEC3B can also be considered a potential target for therapeutic intervention since the inhibition of APOBEC3B catalytic activity has shown a decrease in the rate of tumor mutation and thereby diminishes the chances of unwanted mutation-related consequences, namely metastasis, recurrence, and the occurrence of therapy resistive cancer types [83,84].

The ENAH gene encodes a member of the enabled (Ena)/vasodilator-stimulated phosphoprotein (VASP) family of proteins. It assembles actin filaments for cell motility and adhesion [85]. Recent studies have shown the upregulated expression of ENAH in many human cancer types, namely breast cancer [86,87,88], melanoma [91], pancreatic cancer [92], and gastric cancer [93]. The overexpression of ENAH increases the chances of transformation and tumorigenesis in BC, consequently offering an innovative method of clinical evaluation of BC [94]. The relative expression of hMenaINV and hMena11a splice isoforms of ENAF are valuable biomarkers in the development and progression of human breast carcinoma [95]. In a recent study, the exosomal level of ENAH was significantly higher in BC patients compared to healthy controls [96], thus providing a novel approach for early breast cancer screening.

These concepts elucidate the role of three gene biomarkers, namely MAPK 1, APOBEC3B, and ENAH, in tumorigenesis and potential use as diagnostic and prognostic gene biomarkers of BC. Moreover, these elucidations will significantly influence the therapeutic approaches to managing BC. Thus, our hybrid FS method successfully selected genes that have biological insight and that are potentially druggable targets [94,95,96]

The three best gene biomarker-based ML models were built and tested, and their classification abilities were measured using various statistical metrics. The trained classification models were tested on the independent test data with a balanced distribution of instances in the two classes (the normal and primary breast tumor instances). The XGBoost-based classification model accuracy, F1-score, and AUC value were better than the other classification models, as shown in Table 6. For example, the XGBoost-based model could classify the normal breast samples from the primary breast tumor samples with an accuracy of 0.976 ± 0.027, an F1-Score of 0.974 ± 0.030, and an AUC value of 0.961 ± 0.035 when assessed on an independent test dataset.

The comparative study signifies that the three gene biomarkers obtained using our novel hybrid FS pipeline are highly relevant in classifying primary breast tumors with greater accuracy, thereby enabling the use of the current biomarker-based XGBoost model in the earlier detection of breast cancer. Furthermore, our proposed XGBoost-based web application for classifying primary breast tumors from normal breast gene expression data has been successfully implemented. The web application is available on Render at https://appbcgene.onrender.com/, accessed on 1 December 2022.

## 6. Conclusions and Future Scope

Using novel FS and ML approaches to identify potential gene biomarkers could help classify primary breast tumors. Furthermore, the early detection of BC will help slow down BC’s progression and potentially improve the survival rate through therapeutic interventions at a precise time. To conclude, we propose that, using a robust hybrid FS method, the present study identified a robust and optimal set of three gene biomarkers (MAPK 1, APOBEC3B, and ENAH) for detecting the primary breast tumors of BC patients. Furthermore, the ontology studies show that the gene biomarkers screened using the hybrid approach have biological insight and play an essential role in breast tumorigenesis. Thus, it could be used as BC’s diagnostic and prognostic marker.

Moreover, in the future, applying a different pipeline of hybrid FS approaches and having access to a large sample size gene expression dataset from different institutions (multi-center study) and screening key features from various relevant data sources could be used to develop an enhanced model to classify primary breast tumors from normal breast samples. 

However, the proposed hybrid feature selection framework achieved state-of-the-art accuracy in classifying primary breast tumors from the gene expression data. However, there are still some limitations in the present study. First, the dataset used in the present study is a binary class data, and the current approach may not give the same response as a multiclass dataset. In addition, the proposed framework was applied to a smaller size of training data. Therefore, we need to apply the novel hybrid FS framework to a larger training sample size to evaluate the robustness of the hybrid FS framework in screening the best optimal solution from a given feature space. 

## Figures and Tables

**Figure 1 diagnostics-13-00708-f001:**
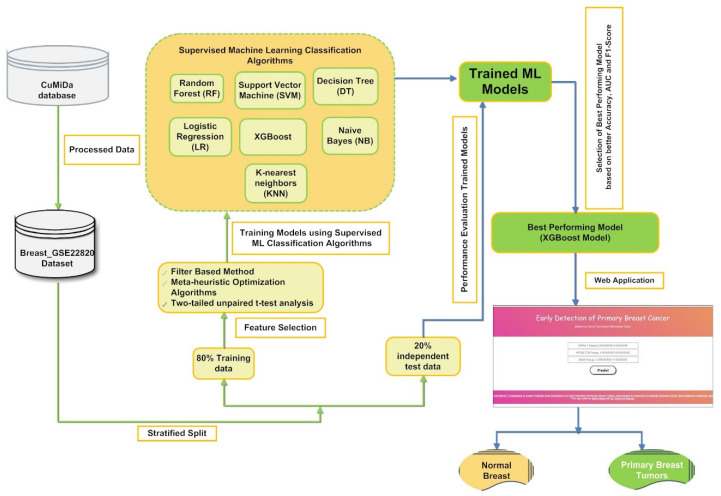
A pictorial representation of the proposed framework for screening a stable set of gene biomarkers from gene microarray data to build a supervised classifier-based application for classifying primary breast cancer samples from normal breast samples.

**Figure 2 diagnostics-13-00708-f002:**
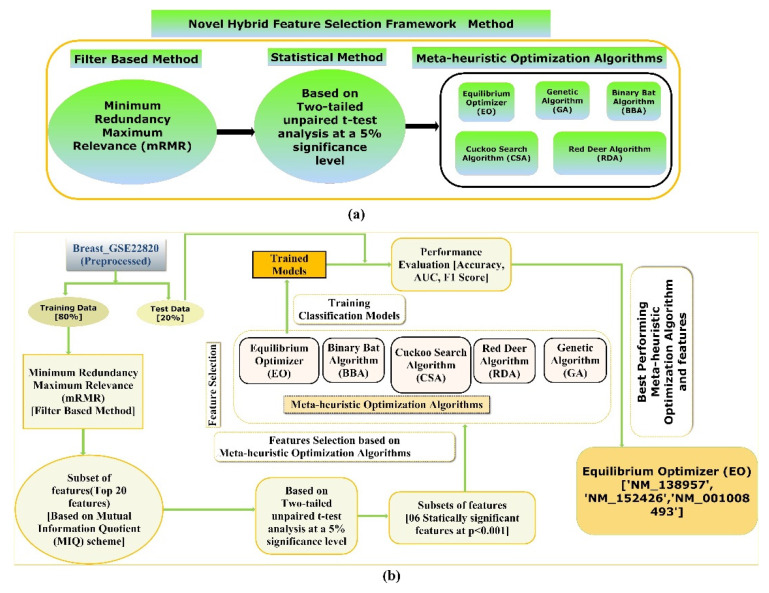
(**a**) An illustration demonstrating the proposed hybrid FS technique and (**b**) a diagram showing the workflow of the proposed method to screen the best hybrid FS methods and an efficient supervised classification-based model for classifying primary breast cancer samples from the normal breast samples.

**Figure 3 diagnostics-13-00708-f003:**
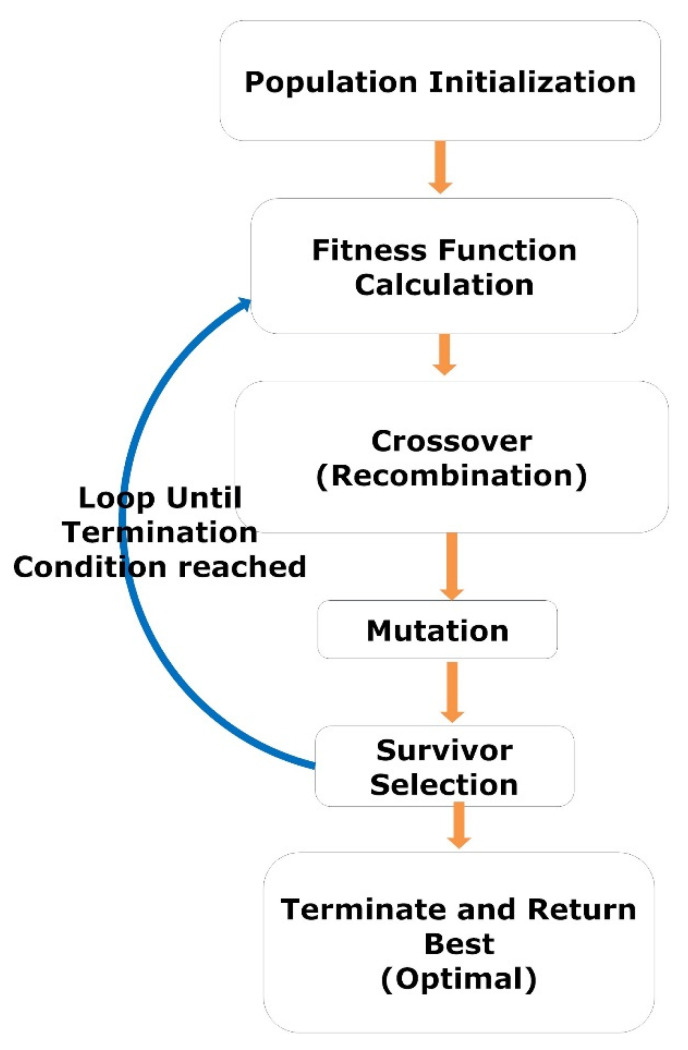
The workflow represents the steps of GA.

**Figure 4 diagnostics-13-00708-f004:**
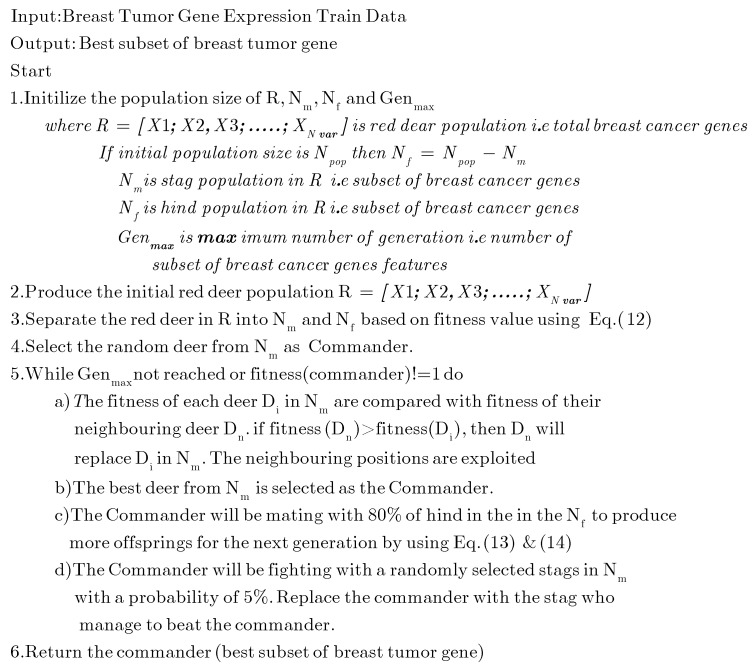
Pseudocode of the Red Deer algorithm.

**Figure 5 diagnostics-13-00708-f005:**
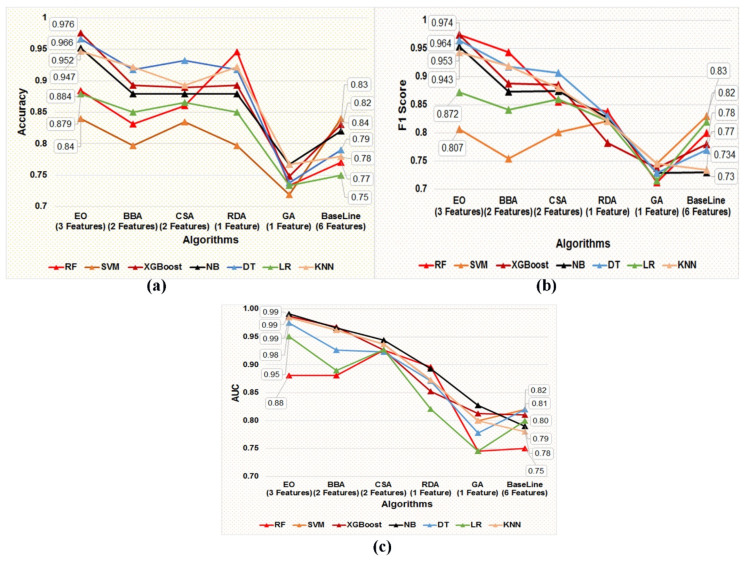
(**a**–**c**) Graphical representation of the comparative performance matrices, namely (**a**) accuracy, (**b**) F1 score, and (**c**) AUC value of the seven classification models developed employing the five optimal subsets of feature(s) screened using five metaheuristic techniques, such as EO, BBA, CSA, RDA, and GA, respectively.

**Figure 6 diagnostics-13-00708-f006:**
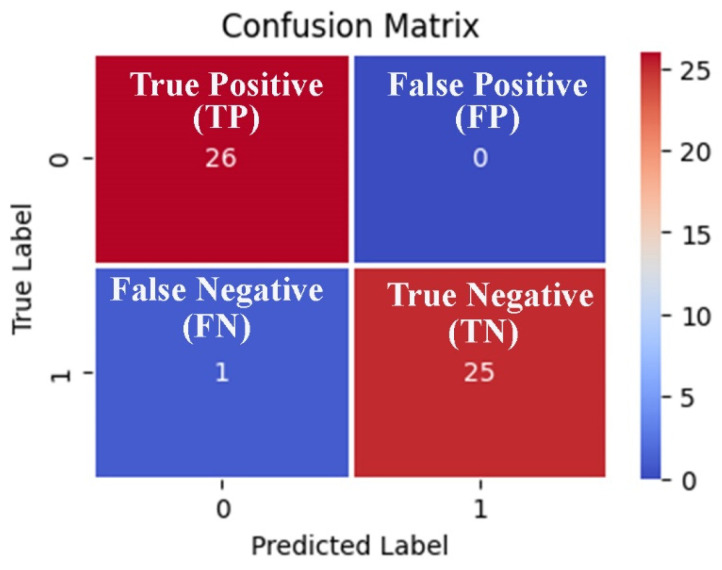
Confusion matrix-based visualization and summarization of the XGBoost-based classification model performance tested on the 20% independent test data generated using the three features obtained from the EO metaheuristic algorithm.

**Figure 7 diagnostics-13-00708-f007:**
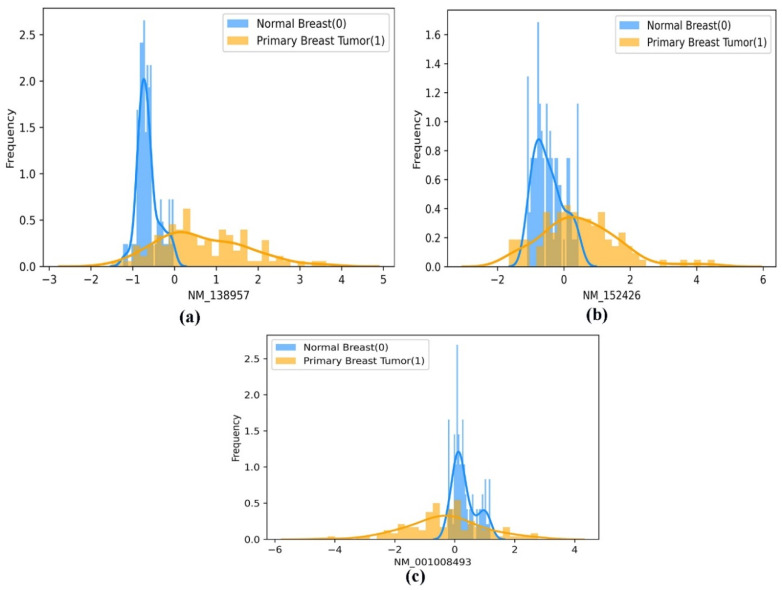
(**a**–**c**). A pictorial representation of the histogram-based frequency distribution of the three optimal gene biomarkers (**a**) ‘NM_138957’, (**b**) ‘NM_152426’, and (**c**) ‘NM_001008493’, between the two classes of population (normal breast and primary breast tumor samples).

**Table 1 diagnostics-13-00708-t001:** Comparison of various FS algorithms for selecting the most informative features to classify microarray BC data. The symbol “-“means the unavailability of data.

Studies Reference Number	Feature Selection Methods			Classification Methods	No. of Selected Genes/miRNA/DNAm	Percentage Accuracy/AUC	Limitations
	Filter Methods	Wrapper Methods	Other Methods				
Shaban et al. [25]	IG	Bat Algorithm	Particle Swarm Optimization (PSO)	NB	5	0.97	Cross-validation studies were not performed for the validity of the results, and there was a higher degree of complexity.
Tahmouresi et al. [25]	Gene Rank	iBGSA	-	SVM	73 genes	84.5/-	The higher number of screened features makes the decision-making system complex and has a smaller sample size.
Hamim et al. [26]	Fisher Score (FS)	ACO	-	C5.0 Decision Tree	Five genes	95.44/0.96	Only one Filter method and wrapper method were used in the study.
Ghozy et al. [27]	IG	GA	-	FLNN	49 genes	85.63/-	A higher number of screened features makes the decision-making system complex, and a smaller sample size and only one filter and wrapper method were used in the study.
AbdElNabi [27]	IG	GWO	-	SVM	70 genes	94.87/-	The higher number of screened features makes the decision-making system complex, and a smaller sample size and only one wrapper method were used in the study for analysis.
Tang et al. [28]	Maximum–Relevance–Maximum–Distance (MRMD)	-	Principal Component Analysis (PCA)	RF	25 (miRNA) 971(CpGs)-MRMD 573 (CpGs)-PCA	91.3/-	A higher number of screened features makes the decision-making system complex and the training time longer, and cross-validation studies were not performed.
Jain et al. [29]	CFS	iBPSO	-	NB	32	92.75/-	The higher number of screened features makes the decision-making system complex, there is a very low sample size, and only one wrapper method is used in the study for analysis.
Shukla et al. [30]	CMIM	AGA	-	ELM	6	94.29/-	The study used a very low sample size and only one filter-based FS and the wrapper-based algorithm for analysis.
Lu et al. [31]	MIM	AGA	-	ELM	216	95.21/-	The higher number of screened features makes the decision-making system complex. There is a very low sample size, and only one filter-based FS algorithm and the wrapper-based algorithm are used in the study for analysis.
Mohapatra et al. [32]	-	MCSO	Max-min scaling/normalization	Kernel Ridge Regression (KRR)	50	97.0/-	A higher number of screened features makes the decision-making system complex, and the study has a low sample size.
Shreem et al. [33]	SU	HS	-	Instance-based Learning algorithm -1 (IB1)	25	83.39/-	A small sample size and a higher number of screened features make the decision-making system complex.

**Table 2 diagnostics-13-00708-t002:** The balanced microarray gene expression data is distributed across the training and testing data.

Class	Training Data	Test Data	Total
Primary Breast Tumor	141	35	176
Normal Breast	141	35	176
Total	282	70	352

**Table 3 diagnostics-13-00708-t003:** Feature selection methods’ parameter settings.

Feature Selection Algorithm	Type of FS Method	Parameter	Value
mRMR	Filter	The minimum value of Correlation	0.00001
		score_func	FCQ
		K [No. of feature to select]	20
		Execution time in CPU	20.4 s
Bat Algorithm	Heuristics	Number of agents	20
		Max_Iteration	100
		Loudness A	1.0
		Emission Rate (r)	0.15
		Alpha	0.95
		Gamma	0.5
		Minimum Frequency	0
		Maximum Frequency	2
		Stopping criteria	Max Iteration
		save_convergence_graph	False
		Execution time in CPU	8.16 s
Genetic Algorithm	Heuristics	Number of agents	20
		Max_Iteration	100
		The objective function (Fitness)	roulette_wheel
		probability of crossover	0.4
		probability of Mutation	0.3
		Stopping criteria	Max Iteration
		save_convergence_graph	False
		Execution time in CPU	12.64 s
Equilibrium Optimizer Algorithm	Heuristics	Number of agents	20
		Max_Iteration	100
		Objective Function	Compute Fitness
		The shape of the transfer function	s
		Pool size	4
		Omega	0.9
		Constant (a2)	1
		weight constant coefficient of the global search (a1)	2
		Generation rate (GP)	0.5
		Stopping criteria	Max Iteration
		save_convergence_graph	False
		Execution time in CPU	4.80 s
Cuckoo Search Algorithm	Heuristics	Number of agents	20
		Max_Iteration	100
		Fraction of nests to be replaced	0.25
		Objective Function	Compute Fitness
		The shape of the transfer function	s
		save_convergence_graph	False
		Stopping criteria	Max Iteration
		Execution time in CPU	7.48 s
Red Deer Algorithm	Heuristics	Number of agents	20
		Max_Iteration	100
		Upper bound	+5
		Lower bound	−5
		Fraction of the total number of males who are chosen as commanders (γ)	0.5
		Fraction of the total number of hinds in a harem who mate with the commander of their harem (α)	0.2
		Fraction of the total number of hinds in a harem who mate with the commander of a different harem (β)	0.1
		Objective Function	Compute Fitness
		The shape of the transfer function	s
		save_convergence_graph	False
		Stopping criteria	Max Iteration
		Execution time in secs	15.99 s

**Table 4 diagnostics-13-00708-t004:** Machine Learning models parameter setting.

Machine Learning Models	Parameter	Value
Logistic Regression	Solver	*lbfgs*
	Penalty	L2
	Regularization strength (C)	1
	tolerance for stopping criteria	0.0001
	Dual	False
	Maximum iteration	100
	Intercept scaling	1
XGBoost	Maximum depth	4
	Learning rate	0.2
	L2 regularization term on weights (re_lambda)	1
	number of boosting rounds (n_estimators)	150
	subsample ratio of the training instance (subsample ratio)	0.9
	subsample ratio of columns when constructing a tree (colsample_bytree)	0.9
	Random number seed (random_state)	1
Gaussian Naïve Bayes	The prior probability of classes	None
	var_smoothing	1 × 10^−9^
Decision Tree	Criterion	Gini
	splitter	best
	maximum depth	4
	min_samples_split	2
	min_samples_leaf	1
	min_weight_fraction_leaf	0.0
	max_features	None
	random_state	None
	max_leaf_nodes	None
	min_impurity_decrease	0.0
	class_weight	None
	ccp_alpha	0.0
K-Nearest Neighbor	N_neighbors	5
	weights	uniform
	Algorithm	auto
	leaf_size	30
	p (power parameter for the Minkowski metric)	Minkowski
Support Vector Machine	Penalty	L2
	Loss function	squared_hinge
	tolerance for stopping criteria (tol)	0.0001
	C (regularization parameter)	1
	fit_intercept	True
	intercept_scaling	1
Random Forest	Criterion	Gini
	n_estimators	100
	Maximum depth	4
	min_samples_split	2
	Min_sample_leafs	1
	Maximum features	auto
	Maximum leaf nodes	none
	min_impurity_decrease	0.0
	bootstrap	True
	Number of trees	90
	oob_score	False

**Table 5 diagnostics-13-00708-t005:** List of the top twenty gene biomarkers from the Breast_GSE22820 gene expression microarray dataset using the FCQ variant of the mRMR FS method.

Sl.no.	Gene Biomarkers
1	NM_152426
2	BC001335
3	BC016934
4	THC2326033
5	A_24_P556328
6	NM_014015
7	BC043603
8	NM_138957
9	A_24_P268474
10	AK027315
11	XM_927487
12	NM_001009185
13	AB033060
14	NM_152426
15	THC2350023
16	THC2438685
17	NM_001008493
18	NM_014674
19	CR606969
20	NM_006579

**Table 6 diagnostics-13-00708-t006:** A listing of the twenty features (Gene biomarkers) attained using mRMR and the resultant mean difference of each twenty features between two classes of the sample population (primary breast cancer and normal breast) at a significance level of 5%.

Sl.no.	Gene Biomarkers	Average and Standard Deviation of Gene Biomarkers between Two Classes of Population (Primary Breast Cancer and Normal Breast)		Unpaired Two-Tailed *t*-Test *p*-Value of the Mean Difference of the Gene Biomarkers between the Two Classes of the Sample Population
		Primary Breast Cancer	Normal Breast	
1	NM_152426	6.8492 ± 1.274	5.9011 ± 0.458	*p* < 0.0001
2	BC001335	6.243 ± 0.568	6.8965 ± 0.256	0.5601
3	BC016934	6.134 ± 0.832	6.383 ± 0.444	*p* < 0.0001
4	THC2326033	7.633 ± 0.590	7.356 ± 0.310	0.0684
5	A_24_P556328	5.795 ± 0.776	6.078 ± 0.669	0.0518
6	NM_014015	11.227 ± 0.422	11.318 ± 0.107	0.0184
7	BC043603	6.006 ± 0.649	5.865 ± 0.455	0.2540
8	NM_138957	7.882 ± 0.572	7.139 ± 0.137	*p* < 0.0001
9	A_24_P268474	6.192 ± 0.554	6.254 ± 0.314	0.2627
10	AK027315	8.121 ± 0.644	7.834 ± 0.644	0.2404
11	XM_927487	5.662 ± 0.389	6.005 ± 0.389	0.1518
12	NM_001009185	5.121 ± 0.960	4.834 ± 0.226	0.3012
13	AB033060	5.410 ± 0.623	4.910 ± 0.284	0.1424
14	NM_004630	12.220 ± 0.353	12.458± 0.125	0.4210
15	THC2350023	6.668 ± 0.918	6.873 ± 0.663	0.3601
16	THC2438685	4.930 ± 0.396	5.386 ± 0.240	0.3684
17	NM_001008493	6.854 ± 0.731	7.064 ± 0.229	0.0010
18	NM_014674	9.236 ± 0.484	9.213 ± 0.263	0.6274
19	CR606969	9.010 ± 0.805	8.997 ± 0.243	0.8556
20	NM_006579	12.413 ± 0.535	11.781 ± 0.227	*p* < 0.0001

**Table 7 diagnostics-13-00708-t007:** The five state-of-the-art metaheuristic algorithms obtained a listing of five optimal global subsets of feature(s).

Metaheuristic Algorithms	Global Optimal Feature(s) Subset
EO	‘NM_138957’, ‘NM_152426’,’NM_001008493’
BBA	NM_152426’, ‘NM_138957’
CSA	BC016934,’NM_006579’
RDA	BC016934
GA	NM_152426

**Table 8 diagnostics-13-00708-t008:** Tabulated results representing the comparative performance evaluation of the seven classification models developed using the five optimal subsets of feature(s) screened using five metaheuristic techniques, namely EO, BBA, CSA, RDA, and GA, respectively.

**Accuracy**							
	**RF**	**SVM**	**XGBoost**	**NB**	**DT**	**LR**	**KNN**
EO	0.884 ± 0.038	0.840 ± 0.045	0.976 ± 0.027	0.952 ± 0.026	0.966 ± 0.025	0.879 ± 0.040	0.947 ± 0.018
BBA	0.831 ± 0.056	0.797 ± 0.066	0.893 ± 0.048	0.879 ± 0.045	0.918 ± 0.033	0.850 ± 0.040	0.922 ± 0.024
CSA	0.860 ± 0.075	0.835 ± 0.042	0.889 ± 0.074	0.879 ± 0.085	0.932 ± 0.052	0.865 ± 0.081	0.893 ± 0.039
RDA	0.946 ± 0.039	0.797 ± 0.066	0.893 ± 0.048	0.879 ± 0.045	0.918 ± 0.033	0.850 ± 0.040	0.922 ± 0.024
GA	0.734 ± 0.073	0.719 ± 0.079	0.748 ± 0.067	0.767 ± 0.071	0.738 ± 0.062	0.734 ± 0.085	0.767 ± 0.048
Baseline	0.77 ± 0.062	0.84 ± 0.058	0.83 ± 0.040	0.82 ± 0.027	0.79 ± 0.047	0.75 ± 0.025	0.78 ± 0.075
**F1 Score**							
	**RF**	**SVM**	**XGBoost**	**NB**	**DT**	**LR**	**KNN**
EO	0.974 ± 0.030	0.807 ± 0.059	0.974 ± 0.030	0.953 ± 0.023	0.964 ± 0.028	0.872 ± 0.047	0.943 ± 0.020
BBA	0.943 ± 0.042	0.754 ± 0.094	0.888 ± 0.053	0.873 ± 0.049	0.918 ± 0.031	0.841 ± 0.040	0.919 ± 0.020
CSA	0.855 ± 0.079	0.801 ± 0.071	0.886 ± 0.075	0.874 ± 0.092	0.907 ± 0.038	0.860 ± 0.085	0.880 ± 0.053
RDA	0.838 ± 0.088	0.821 ± 0.075	0.782 ± 0.079	0.827 ± 0.093	0.830 ± 0.098	0.821 ± 0.100	0.821 ± 0.075
GA	0.712 ± 0.077	0.745 ± 0.062	0.738 ± 0.076	0.729 ± 0.087	0.729 ± 0.068	0.715 ± 0.087	0.745 ± 0.062
Baseline	0.80 ± 0.034	0.83 ± 0.052	0.78 ± 0.036	0.73 ± 0.062	0.77 ± 0.035	0.82 ± 0.027	0.734 ± 0.073
**AUC Value**							
	**RF**	**SVM**	**XGBoost**	**NB**	**DT**	**LR**	**KNN**
EO	0.881 ± 0.038	0.985 ± 0.020	0.987 ± 0.025	0.991 ± 0.017	0.975 ± 0.016	0.951 ± 0.035	0.985 ± 0.020
BBA	0.881 ± 0.040	0.962 ± 0.026	0.968 ±0.022	0.966 ± 0.022	0.926 ± 0.026	0.890 ± 0.035	0.962 ± 0.026
CSA	0.926 ± 0.062	0.937 ± 0.049	0.926 ± 0.055	0.944 ± 0.048	0.923 ± 0.056	0.927 ± 0.060	0.937 ± 0.049
RDA	0.896 ± 0.066	0.873 ± 0.055	0.852 ± 0.071	0.893 ± 0.062	0.871 ± 0.056	0.821 ± 0.100	0.873 ± 0.055
GA	0.745 ± 0.085	0.800 ± 0.062	0.813 ± 0.075	0.827 ± 0.101	0.778 ± 0.050	0.745 ± 0.085	0.800 ± 0.062
Baseline	0.75 ± 0.025	0.82 ± 0.027	0.81 ± 0.027	0.79 ± 0.047	0.82 ± 0.027	0.80 ± 0.026	0.78 ± 0.075

**Table 9 diagnostics-13-00708-t009:** Comparative accuracy and AUC value evaluation between models built using features (gene biomarkers) obtained using our proposed hybrid FS method.

**Sl.no.**	**Author**	**Machine Learning Model**	**Hybrid FS Method**	**Gene Biomarker (Features)**	**Accuracy (%)**	**AUC Value**
1	[24]	RF	MRMD + PCA	25 (miRNA) 971 (CpGs)-MRMD 573 (CpGs)-PCA	91.3	-
2	[14]	SVM	Gene Rank + iBGSA	73 genes	84.5	-
3	[15]	C5.0 Decision Tree	Fisher Score + ACO	5 genes	95.44	0.96
4	[27]	FLNN	IG + GA	49 genes	85.63	-
5	[28]	SVM	IG + GWO	70 genes	94.87	-
6	[29]	NB	CFS + iBPSO	32 genes	92.75	-
7	[30]	ELM	CMIM + AGA	6 genes	94.29	-
8	[31]	ELM	MIM + AGA	216 genes	95.21	-
9	[32]	KRR	MCSO + Max-min scaling/normalization	50 genes	97.0	-
10	[33]	IB1	SU + HS	25 genes	83.39	-
11	Our Proposed Classification model	XGBoost	mRMR + Unpaired *t*-test + EO	MAPK1, APOBEC3B, and ENAH	0.976 ± 0.027	0.987 ± 0.025

## Data Availability

Datasets are publicly available at: https://sbcb.inf.ufrgs.br/cumida, accessed on 1 June 2022.

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
