# Peer review of "A Hybrid Machine Learning Approach to Screen Optimal Predictors for the Classification of Primary Breast Tumors from Gene Expression Microarray Data"

_diagnostics, 2023, doi:10.3390/diagnostics13040708_

Round 1
Reviewer 1 Report
An interesting and well presented manuscript on the use of FS for ML computational diganosis.
A small detail to address:
Feature selection "FS" used in line 13 in the abstract, but not yet defined previously in the manuscript.
In clinican applications, TP and FS are very important for diagnosis significance. Still is not commonly used in the computer science field as shown in table 1.
A question to consider. Health and disease state do not display a normal statistical distribution. There's usually less disease data or counts. How does this could affect the FS algorithm or ML classification?
Author Response
Response to Reviewer 1 Comments
Point 1: Feature selection "FS" used in line 13 in the abstract, but not yet defined previously in the manuscript.
Response 1: The comment has been resolved as suggested by the reviewer (Refer to line 13 in the Revised manuscript).
Point 2: In clinican applications, TP and FS are very important for diagnosis significance. Still is not commonly used in the computer science field as shown in table 1.
Response 2: Yes, we agree with the reviewer that TP and FS are significant for diagnosis. In Table 1, we have compared the FS method applied for screening the best-performing predictors for accurate classification and detection of breast cancer from the microarray gene expression dataset. Additionally, we have also compared the accuracy achieved by using different FS methods where the TP plays a significant role in determining the efficacy (accuracy) of any FS method.
Point 3: A question to consider. Health and disease state do not display a normal statistical distribution. There's usually less disease data or counts. How could this affect the FS algorithm or ML classification?
Response 3: We agree with the reviewer that there is usually a lower number of disease samples or counts. The problem with training the model with an imbalanced dataset is that the model will be biased towards the majority class only. This causes a problem when we are interested in predicting the minority class (Like in the present study, where we perform classification of breast cancer from microarray gene expression data. Here, cancer is more important than not having cancer).
Therefore, the model evaluator parameters, such as accuracy and ROC_AUC score, give false results for imbalanced datasets. Most of the classification algorithms, such as KNN, Naïve Bayes, Linear Regression, Logistic Regression, Support Vector Machine, and Decision Tree, get impacted by imbalanced datasets. Therefore, to improve the performance, we need to convert imbalanced datasets into balanced datasets before feature selection (Pes, 2020, Zhang et al., 2017, Luque et al., 2019) using oversampling and undersampling techniques, such as SMOTE, ROS, and RUS. Otherwise, we can use 'Precision/'Recall' performance measures for evaluating models built using imbalanced datasets.
References:
- Pes, B. (2020). Learning from high-dimensional biomedical datasets: the issue of class imbalance. IEEE Access, 8, 13527-13540.
- Zhang, C., Bi, J., & Soda, P. (2017, November). Feature selection and resampling in class imbalance learning: Which comes first? An empirical study in the biological domain. In 2017 IEEE international conference on bioinformatics and biomedicine (BIBM) (pp. 933-938). IEEE.
- Luque, A., Carrasco, A., Martín, A., & de Las Heras, A. (2019). The impact of class imbalance in classification performance metrics based on the binary confusion matrix. Pattern Recognition, 91, 216-231.
Reviewer 2 Report
This paper studies the A Hybrid Machine Learning Approach to Screen Optimal Predictors for the classification of Primary Breast Tumor from Gene Expression Microarray Data. Some weaknesses should be addressed in this paper. Therefore, I suggest the authors resubmit it after a major revision. My suggestions are as follows:
1. Discuss the study's limitations and future research suggestions.
2. I strongly suggest that the paper be proofread and reread meticulously again, particularly regarding the spelling and grammatical mistakes.
3. In Fig1, you provided a flowchart to explain the Graphical abstract of the designed hybrid soft computing approach. This section must provide a concise and clear explanation of the suggested approach. Although the flowchart is beneficial, it’s also important to outline the methodology behind this new approach.
4. Divide the introduction into two parts of introduction and literature review. Add a literature review section after the introduction
5. Please outline the structure of your paper at the end of the introduction with more details.
6. I suggest that you update section 2.1 so that the illustration used in the Machine Learning framework subsection should be more readable.
7. Please clarify the definitions for equations 3 and 4. What is the main strategy behind these two targets?
8. It is necessary to include additional information for the RD algorithm. We have other novel suggested algorithms such as Harris Hawks Optimizer and Lion optimization metaheuristics population-based algorithms. What is the superiority of this algorithm compared with other metaheuristics?
9. Following the mathematical model is difficult due to a few notational mistakes.
10. What is your mean by line 451? "xi" edit mathematical errors.
Please explain and clarify more. Which DMUs? please provide more information
11. To improve your related works, remove at least 10 unrelated references and consider the following hybrid machine learning references.
- A hybrid machine learning approach to network anomaly detection. Information Sciences. 2007 Sep 15;177(18):3799-821.
- Predicting permeability of tight carbonates using a hybrid machine learning approach of modified equilibrium optimizer and extreme learning machine. Acta Geotechnica. 2022 Apr;17(4):1239-55.
-. A novel machine learning approach combined with optimization models for eco-efficiency evaluation. Applied Sciences. 2020 Jul 28;10(15):5210.
- Matchmaking in reward-based crowdfunding platforms: A hybrid machine learning approach. International Journal of Production Research. 2022 Dec 17;60(24):7551-71.
- Artificial intelligence-driven prediction modeling and decision making in spine surgery using hybrid machine learning models. Journal of Personalized Medicine. 2022 Mar 22;12(4):509.
- Novel Enhanced-Grey Wolf Optimization hybrid machine learning technique for biomedical data computation. Computers and Electrical Engineering. 2022 Apr 1;99:107778.
- A Novel Hybrid Machine Learning Based System to Classify Shoulder Implant Manufacturers. InHealthcare 2022 Mar 20 (Vol. 10, No. 3, p. 580). MDPI.
- Developing a novel integrated generalised data envelopment analysis (DEA) to evaluate hospitals providing stroke care services. Bioengineering. 2021 Dec 10;8(12):207.
- Effective forecasting of key features in hospital emergency department: Hybrid deep learning-driven methods. Machine Learning with Applications. 2022 Mar 15;7:100200.
- A novel hybrid parametric and non-parametric optimisation model for average technical efficiency assessment in public hospitals during and post-COVID-19 pandemic. Bioengineering. 2021 Dec 27;9(1):7.
- Evaluation of green logistics efficiency in Jiangxi Province based on Three-Stage DEA from the perspective of high-quality development. Sustainability. 2022 Jan 11;14(2):797.
- An integrated artificial intelligence model for efficiency assessment in pharmaceutical companies during the COVID-19 pandemic. Sustainable Operations and Computers. 2022 Jan 1;3:156-67.
In conclusion, this version is unacceptable and needs to apply all the suggested comments point by point. In particular, applying the suggested high-quality related references
Author Response
Response to Reviewer 2 Comments
Point 1: Discuss the study's limitations and future research suggestions.
Response 1: The limitations and future research of the proposed method have been provided in the revised manuscript as suggested by the reviewer (Refer to lines 976 to 983 for limitations of the study and lines 971 to 975 for future scope in the revised manuscript).
Point 2: I strongly suggest that the paper be proofread and reread meticulously again, particularly regarding the spelling and grammatical mistakes.
Response 2: The manuscript has been edited and proofread using the professional Grammarly Premium English editing software for rectifying spelling and grammatical mistakes as suggested by the reviewer.
Point 3: In Fig1, you provided a flowchart to explain the Graphical abstract of the designed hybrid soft computing approach. This section must provide a concise and clear explanation of the suggested approach. Although the flowchart is beneficial, it’s also important to outline the methodology behind this new approach.
Response 3: A concise and clear explanation of the suggested approach has been provided in lines 128 to 150 of the revised manuscript.
Point 4: Divide the introduction into two parts of introduction and literature review. Add a literature review section after the introduction.
Response 4: The introduction has been divided into two parts (introduction and literature review). The review section has been placed after the introduction as suggested by the reviewer (Refer to lines 29 to 166 for the introduction and 167 to 226 for the literature review).
Point 5: Please outline the structure of your paper at the end of the introduction with more details.
Response 5: A detailed outline of the structure of our paper has been provided at the end of the introduction section of the revised manuscript (Refer to lines 156 to 166).
Point 6: I suggest that you update section 2.1 so that the illustration used in the Machine Learning framework subsection should be more readable.
Response 6: Section 2.1 was edited to make the workflow more readable and has been shifted to the introduction section (Refer to lines 128 to 150). Section 2.1 was shifted to the introduction due to its resemblance with an explanation of the suggested approach (Represented using Figure 1) available in the original manuscript from lines 149 to 153.
Point 7: Please clarify the definitions for equations 3 and 4. What is the main strategy behind these two targets?
Response 7: The strategy behind equation 3 and 4 have been explained as suggested by the reviewer (Refer to lines 315 to 331).
Point 8: It is necessary to include additional information for the RD algorithm. We have other novel suggested algorithms such as Harris Hawks Optimizer and Lion optimization metaheuristics population-based algorithms. What is the superiority of this algorithm compared with other metaheuristics?
Response 8: The RD algorithm has been modified by adding a related equation and a pseudocode to explain the working of the algorithm (Refer to lines 437 to 482). To compare the performance or superiority of the RD algorithm with other metaheuristics algorithm is out of the scope of the current study. However, we have added a sentence in our revised manuscript stating that the RD algorithm combines the important principles of both evolutionary algorithms and heuristics. Thereby providing a reasonable answer to its competence in exploration and exploitation to solve problems in different fields, such as business, research, or industrial applications. However, the detailed comparative performance of the RD algorithm with other algorithms is highlighted in the manuscript, whose link is provided below:
Zitar, R. A., Abualigah, L., & Al-Dmour, N. A. (2021). Review and analysis for the Red Deer Algorithm. Journal of ambient intelligence and humanized computing, 1–11. Advance online publication. https://doi.org/10.1007/s12652-021-03602-1
Here, in the above paper, the algorithm has been compared to the metaheuristic algorithm published in 2021. They found that the RDA proved competitive to comparative metaheuristics methods when applied to optimize different types of benchmark functions. These benchmark functions had different modes, with multiple variables and many local minima. The standard F test showed that the RDA is ranked 4 among the comparative algorithms. Most of those algorithms are well-known and frequently used in different applications. Most of those algorithms, if not all, are metaheuristic-based algorithms that could be swarm intelligence-based, local search-based, physical-based, chemical-based, human-based, and others. The RDA proved its presence among them. The RDA showed exploration and exploitation capabilities based on its procedure, behavior, and simulation results.
The performance of algorithms, namely Harris Hawks Optimizer and Lion optimization, is not involved in the above study. However, a current study by Abdulaziz Alorf in 2023 has shown a comparative performance of all metaheuristics in terms of unimodal function, multimodal function, CEC-BC-2017 functions, and engineering problems. The link to the paper is provided below:
https://www.sciencedirect.com/science/article/pii/S0952197622006121
Point 9: Following the mathematical model is difficult due to a few notational mistakes.
Response 9: As suggested by the reviewer, the notational mistakes in the manuscript have been identified and rectified in the revised manuscript.
Point 10: What is your mean by line 451? "xi" edit mathematical errors. Please explain and clarify more. Which DMUs? please provide more information
Response 10: In lines 533 to 535, the authors want to state that “xi” is an instance of the training dataset, and “yi” represent the target-dependent variable label. Regarding the DMU, we are unclear on what the reviewer means by DMUs. If the reviewer means Decision Making Unit, then we are not sure in what context the reviewer wants to have more explanation and clarification of DMUs and exactly of which DMUs. As we have used no such DMU terms in the manuscript.
Point 11: To improve your related works, remove at least 10 unrelated references and consider the following hybrid machine learning references.
Response 11: We would like to have the kind attention of the reviewer that the authors in the manuscript have worked on building a hybrid framework of feature selection technique and finally evaluating the performance of the best subsets of optimal features to classify primary breast cancer tumor samples using various state-of-the-art machine learning models. Most of the references cited by the reviewer are related to the hybrid machine learning model where no such hybrid feature selection approach was developed by combining different feature selection methods to form a hybrid FS framework to screen breast cancer. Therefore, we have not removed references from our literature review. However, we have added the reference cited by the reviewer in a separate paragraph in the introduction section in lines 90 to 93 of the revised manuscript.
Reviewer 3 Report
In the present article, the authors proposed a hybrid FS sequential framework involving minimum 13 Redundancy-Maximum Relevance (mRMR), a two-tailed unpaired t-test, and meta-heuristics to 14 screen the most optimal set of gene biomarkers as predictors for Breast Cancers. The manuscript is well-written and attracts readers. However, authors are advised to incorporate the following suggestions:
1) Though an extensive study has been provided in the introduction section, authors are required to provide a separate introduction and literature review section.
2) Table 1 provides the FS methods proposed by various authors. Limitations of presented FS can be provided by authors for better understanding.
3) References on the detection of cancers using machine learning techniques can be added to strengthen the literature review section:
a) doi.org/10.1016/j.bspc.2022.103596;
b)https://doi.org/10.1007/s12553-019-00375-8; c) https://doi.org/10.1007/s12652-021-03256-z
4) Page 6, line 170, "the preprocessed data was introduced into a hybrid framework of FS methods which involves a sequential implementation of a minimal-optimal filter-based FS algorithm, a statistical filter, and finally, a set of five state-of-the-art meta-heuristic al algorithms". Here, the authors also need to comment on the execution time of the proposed hybrid FS.
5) Page 6, line 200, " dataset variables to screen and filter similar variables using a variance threshold of 0.01." Since the selection of value of variance threshold plays a very crucial role in selecting the number of features, how the value 0.01 is selected?
6) Since the dataset is highly unbalanced (1:17.6), the authors proposed SMOTE to tackle the problem. Authors need to provide the simulation parameters for SMOTE for the reproducibility of results.
7) It was noted from figure 2 that the authors utilized hybrid FS on the training dataset. The ML models are trained with a reduced feature dataset. However, for testing purposes the authors employed a dataset (with all features), since the ML models are trained on a reduced feature dataset then definitely the behavior of ML models will show unfavorable results. Justify it.
8) The authors need to provide simulation parameters for all the FS, ML models etc. for result reproducibility.
9) For a better understanding of results, authors need to specify the total number of samples after the implementation of SMOTE, the number of samples for training and testing of ML models.
10) The authors need to clearly specify the number of features selected by each FS ( presently only provide only top 20, 05% etc.) i.e sections 3.1.1 to 3.1.3 needs to be elaborated.
11) Provide scope of improvement in proposed work in conclusion section.
how effectively the trained models s
Author Response
Response to Reviewer 3 Comments
Point 1: Though an extensive study has been provided in the introduction section, authors are required to provide a separate introduction and literature review section.
Response 1: The introduction has been divided into two parts (introduction and literature review). The review section has been placed after the introduction as suggested by the reviewer (Refer to lines 29 to 166 for the introduction and 167 to 223 for the literature review).
Point 2: Table 1 provides the FS methods proposed by various authors. Limitations of presented FS can be provided by authors for better understanding.
Response 2: As suggested by the reviewer, the limitations of the presented FS algorithms have been provided by the authors in Table 1 (Refer to lines 225 to 226)
Point 3: References on the detection of cancers using machine learning techniques can be added to strengthen the literature review section:
- a) doi.org/10.1016/j.bspc.2022.103596;
b)https://doi.org/10.1007/s12553-019-00375-8; c) https://doi.org/10.1007/s12652-021-03256-z.
Response 3: We would like to have the kind attention of the reviewer that the authors in the manuscript have worked on building a hybrid framework of feature selection technique and finally evaluating the performance of the best subsets of optimal features to classify primary breast cancer tumor samples using various state-of-the-art machine learning models. The reference cited by the reviewer is related to the hybrid machine learning model, where the hybrid feature selection approach was developed for screening different cancer but not breast cancer. Therefore, we have not added these references to our literature review. However, we have added the reference cited by the reviewer in a separate paragraph in the introduction section in line number 93 of the revised manuscript.
Point 4: Page 6, line 170, "the preprocessed data was introduced into a hybrid framework of FS methods which involves a sequential implementation of a minimal-optimal filter-based FS algorithm, a statistical filter, and finally, a set of five state-of-the-art meta-heuristic al algorithms". Here, the authors also need to comment on the execution time of the proposed hybrid FS.
Response 4: The execution time of the proposed hybrid FS framework has been added in Table 3 as suggested by the reviewer (Refer to lines 487 of the revised manuscript).
Point 5: Page 6, line 200, " dataset variables to screen and filter similar variables using a variance threshold of 0.01." Since the selection of value of variance threshold plays a very crucial role in selecting the number of features, how the value 0.01 is selected?
Response 5: As a rule of thumb, we selected the default value of 0.01 as the threshold to remove the quasi-constant features (features that are almost constant) that have more than 99% similar values for the output observations.
Point 6: Since the dataset is highly unbalanced (1:17.6), the authors proposed SMOTE to tackle the problem. Authors need to provide the simulation parameters for SMOTE for the reproducibility of results.
Response 6: The simulation parameters for SMOTE have been provided as suggested by the reviewer (Refer to lines 250-252 for the revised manuscript).
Point 7: It was noted from figure 2 that the authors utilized hybrid FS on the training dataset. The ML models are trained with a reduced feature dataset. However, for testing purposes the authors employed a dataset (with all features), since the ML models are trained on a reduced feature dataset then definitely the behavior of ML models will show unfavorable results. Justify it.
Response 7: The author would like to bring to your notice that, in the present study, the ML models were trained and tested with the reduced feature dataset as stated in section 3.4 in lines 489 to 508, section 4.2 in lines 790 to 795 and also in the introduction section from line number 137 to 146.
Point 8: The authors need to provide simulation parameters for all the FS, ML models etc. for result reproducibility.
Response 8: The simulation parameters for all the FS, ML models, etc., for result reproducibility, have been provided by the authors as suggested by the reviewer (Refer to Table 3, lines 487 to 488 and Table 4, lines 635 to 636 in the revised manuscript).
Point 9: For a better understanding of results, authors need to specify the total number of samples after the implementation of SMOTE, the number of samples for training and testing of ML models.
Response 9: As suggested by the reviewer, the number of samples for training and testing of ML models have been tabulated in Table 2 in the revised manuscript (Refer to lines 254 to 260).
Point 10: The authors need to clearly specify the number of features selected by each FS ( presently only provide only top 20, 05% etc.) i.e sections 3.1.1 to 3.1.3 needs to be elaborated.
Response 10: The number of features selected using the three-feature selection method of the hybrid framework is listed in section 4.1.1. to 4.1.3 in lines 747- 752 (mRMR feature selection method), 764- 771 (unpaired t-test), and 774 to 785 (five metaheuristics algorithms), respectively.
Point 11: Provide scope of improvement in proposed work in conclusion section.
Response 11: The future scope of the present study has been added and stated in the conclusion section of the manuscript (Refer to lines 971 to 975).
Round 2
Reviewer 2 Report
Authors answered all my concerns and this version is acceptable for publication.
Reviewer 3 Report
The authors put significant efforts to address the comments.
All my previous comments were addressed and the manuscript can be accepted.